# Learning to Rank for In-Context Example Retrieval

**Yuwen Ji,**[1,2*]   **Luodan Zhang,**[2*]   **Ambyer Han,**[3*]
**Haoran Que,**[4]   **Lei Shi,**[5]   **Wang Chao,**[3]   **Yue Zhang**[2†]

[1]Zhejiang University   [2]Westlake University
[3]Amap, Alibaba Group   [4]Peking University   [5]Beihang University
{zhangluodan, zhangyue, jiyuwen}@westlake.edu.cn

## Abstract

Recent advances in retrieval-based in-context learning (ICL) train the retriever using a classification objective, which categorizes in-context examples (ICEs) into the most useful and the rest based on absolute scores. However, during inference, ICEs are *retrieved by score ranking* rather than classification — The classification training objective deviates from this test scenario. Hence, in this paper, we propose a novel algorithm that trains a retrieval model by ranking formulation, where the preference rankings between ICEs are given by comparing the likelihood of the LLM generating the correct answer conditioned on each exemplar. By learning to rank, we motivate the retriever to automatically learn diverse rationales why specific examples are more useful for ICL decisions. This addresses the issue that classification models poorly capture broader utility. Experimental results demonstrate the top-1 performance of our proposal across 9 NLP tasks, with ablation studies and case studies further validating the effectiveness of our design. *The code can be found in:* `https://github.com/2022neo/SeDPO_NIPS25`

## 1   Introduction

Large Language Models (LLMs) [1, 2, 3] have shown their versatility in addressing diverse problems through in-context learning (ICL), which can be viewed as few-shot learning. ICL [4, 5, 6, 7] allows providing a few in-context examples (ICEs) to guide LLMs in generating predictions for test inputs without parameter updates. When there is a large set of labeled data, selecting the most useful few examples can improve ICL performance. To this end, existing methods [8, 9] fine-tune LMs as dense retrievers, typically in two steps: (1) Scoring a set of examples one by one or group by group using the ICL LLM; (2) Training the retriever model on scored data to align with the ICL LLM.

Dominant approaches train the retriever using a classification objective [10], categorizing ICEs into the most useful and the rest based on absolute scores. However, during inference, ICEs are *retrieved by score ranking* rather than classification — The classification training objective deviates from this test scenario, which leads to limitations. Taking math problems as an example, when there are no reference answers for a test query, recommending other useful ICEs, e.g., relevant formulas, will be more valuable. What we care about ultimately is not the absolute classification of examples, but the relative orders conditioned by the test input — which sets of examples are more useful.

We formulate in-context example retrieval as an information retrieval (IR) task by adopting a learning-to-rank (LTR) objective. LTR has been widely used in information retrieval [11], naturally capturing the broader utility across examples with ranking formulation. However, it remains underexplored in

---

*Equal contribution.

†Corresponding author.

39th Conference on Neural Information Processing Systems (NeurIPS 2025).

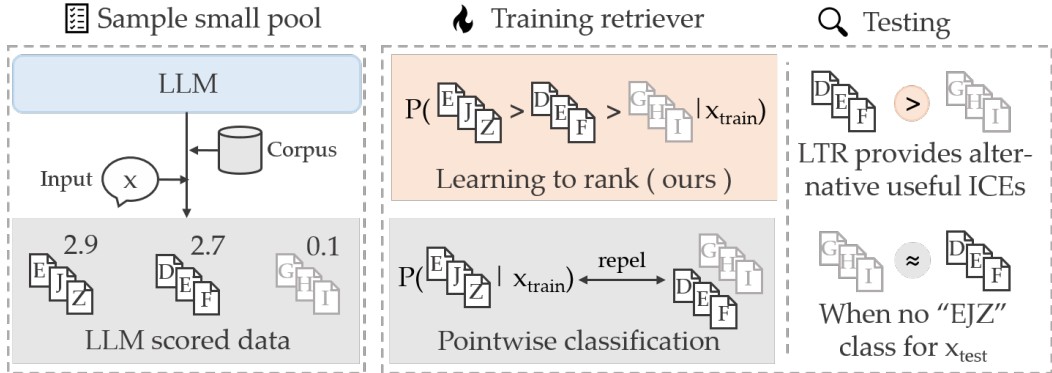

Figure 1: Motivating framework of existing work and our proposal (in red). E.g., when there are no reference answers (EJZ) for test input, LTR recommends alternative concepts or formulas (DEF).

ICL. As depicted in Figure 1, using partial order knowledge between scored examples, a retriever can retrieve a broader range of useful examples with ranking formulation, improving ICL performance.

To train the retriever, we propose a novel algorithm that aligns the preference ranking of ICEs, by comparing the likelihood of the LLM generating the correct answer conditioned on each exemplar. By LTR, we motivate the retriever to automatically learn various rationales why ICEs are more useful for ICL decisions. This addresses the issue that classification models poorly capture broader utility. To learn how to rank training examples, we adapt direct preference optimization [12] (**DPO**), integrating the **S**equential **E**xample relaxation [13], so as to derive a trainable pairwise ranking objective, while implicitly conforming to global preference order constraints. We thus name this algorithm **SeDPO**.

Experimentally, our method constantly ranks in top-1 across 9 NLP tasks, with an improvement of up to 18% over the best-performing classification-based baseline, achieving the SOTA results. Ablation study further confirms the usefulness of our ranking formulation and its complementary strength to existing paradigms. We summarize our key contributions as follows:

- We propose to learn preference ranking orders for ICL example retrieval.
- We introduce an extension of DPO for training retriever with pairwise ranking formulation.
- We demonstrate that SeDPO significantly outperforms existing state-of-the-art retrieval methods for in-context example retrieval.

## 2 Related work

**Retrieval-based ICL.** Two primary types of retrievers are commonly used for sample selection in ICL. The first type consists of off-the-shelf retrievers derived from heuristic criteria [14, 15, 16, 17, 18, 19, 20, 21, 22, 23, 24, 25]. However, the assumption that model performance is always correlated with heuristic criteria is not reliable. Consequently, other approaches train a retriever on a corpus using LLM's feedback so as to select ICEs that the LLM truly prefers. EPR [26] encodes and selects few-shot examples independently, with both queries and examples being encoded as vectors, enabling rapid retrieval through DPR (dense passage retrieval) [27] during inference. Following EPR's paradigm, CEIL [28] and UPRISE [29] are proposed to tackle various aspects of NLP tasks.

Observing that the above methods can overlook the interaction of examples that are used together as one few-shot prompt, the latest advancements achieve the SOTA performance by modeling the sequential order between examples in each prompt, such as RetICL [30], Active Example Selection [31], and $Se^2$ [13]. Among these, RetICL and Active Example Selection formalize the problem as Markov Decision Processes (MDP) and optimize models through reinforcement learning (RL). However, they suffer from training instability. In contrast, $Se^2$ models sequential order among examples in each prompt directly on the input text, achieving closed-loop optimization and the SOTA performance, making it the most suitable baseline in this class. Neither of these methods considers ranking orders of examples across prompts, which we address in this paper.

Recent ICL papers have considered aspects that are complementary to our work: (1) BESC [32] focuses on the internal ordering of ICEs within each prompt, using a contrastive loss to incrementally construct optimal sequences with dynamic lengths step-by-step. By contrast, our work emphasizes the ranking of different prompts. (2) CASE [33] prioritizes the efficiency of the selection process, framing it as a "top-m arm identification" problem with absolute training rewards. We, however, prioritize the quality of retrieved examples, formulating selection as a "learning to rank" problem with pairwise training rewards. (3) CLG [34] is a task-level selection method for few-shot scenarios, where scalability is critical—it selects a fixed set of examples for all test queries via gradient matching. Our work, by contrast, is optimized for few-shot scenarios where query-specific utility of each example is essential. We leave the exploration of these complementary aspects for future work.

**Preference-aware ICL.** Example retrieval methods considering the preference ranking orders of ICL examples are limited. UDR [35] additionally fits mini-batch rank indices with LambdaRank loss [36], which may conflict with global rank indices, as inserting a new scored example between two originally scored examples alters the local rank indices. This discrepancy amplifies as the example pool scales. Another line is RL-ICL [37], which develops a self-retrievable LLM with PPO [38] by modeling ICL performance as a reward signal. Due to differences in experimental and retrieval setups, we focus our comparison on mainstream retriever-based methods. A few ICL methods [39, 40] consider the preference order as an evaluation metric. In contrast, our method can efficiently learn the ranking orders of scored examples over the entire corpus.

## 3 Method

We first formalize the problem of ICL for LLM (Section 3.1), and subsequently use it to label ranking data (Section 3.2). Finally, we use the ranked data to train one different retriever (Section 3.3).

### 3.1 In-context learning with sequential example retrieval

**In-context learning** [41] is a key capability of LLMs where the model learns from a few examples provided in the input prompt to perform a task without parameter updates. Following the previous definition [13, 30], given a test sample $(x, y)$, the LLM predicts $\hat{y}$ based on examples and input $x$ as:

$$\hat{y} = \text{LLM}(e_K \oplus e_{K-1} \oplus ... \oplus e_1 \oplus x), \tag{1}$$

where $e_k = (x_k, y_k)_{k=1}^{K}$ is an example drawn from a corpus $\mathcal{C}$, consisting of an input-output pair. $K$ is the shot number and $\oplus$ is the concatenation. The retrieval objective is to seek a set of examples in $\mathcal{C}$ for test input $x$, putting them into a sequential order $(e_1, ..., e_{K-1}, e_K) = e_{[1:K]}$ following [42], aiming to make $\hat{y}$ match the label $y$. All candidate $K$-shot sequences are denoted as $\mathcal{E} = \{e_{[1:K]}\}$. Notably, we specify in Eq. (1) that ordered examples are input right-to-left. As long as the ICL templates remain consistent during training and inference, the left-to-right input order is also supported.

**Retriever model.** We consider a class of methods based on dense passage retrieval (DPR) as our retriever model. DPR consists of a query encoder $\text{E}_{\text{input}}(\cdot)$ and an example encoder $\text{E}_{\text{example}}(\cdot)$, often initialized with a pretrained text-encoder such as BERT-base-uncased [43]. The retrieval score $\phi_{\text{retriever}}$ of example $e$ for test input $x$ conditioned on $c$ is computed as $\text{sim}(e, x|c) = \text{E}_{\text{example}}(e)^{\top} \text{E}_{\text{input}}(c \oplus x)$.

**Retrieval process.** Following [13], we use the same beam search during inference to retrieve $e_{[1:K]}$ for fair comparison. We first encode and index all examples $e_k \in \mathcal{C}$ using the trained $\text{E}_{\text{example}}(\cdot)$. Given a test input $x$, we encode $x$ with $\text{E}_{\text{input}}(\cdot)$ and retrieve $w$ examples with the highest retrieval scores $\text{E}_{\text{example}}(\cdot)^{\top} \text{E}_{\text{input}}(\cdot)$. We set default beam size $w = 3$, the same as in [13]. We also draw $w = 1$ for brief illustration in Figure 2(c). These examples are then concatenated with the current inputs as new context sequences. The retriever scores are accumulated into the sequences' scores. This process is repeated, encoding inputs and seeking examples to maintain the $w$ highest scoring candidate sequences until each sequence contains K examples. The sequence with highest accumulated scores is chosen as $e_{[1:K]}$. Motivated by left-to-right generation in autoregressive models, this retrieval process lets later retrieved ICEs observe previous retrieved ones, being aware of sequential relationships.

**Training task.** To convert LTR for ICL into a formal goal, we design our training task as follows: (1) A scoring function $\phi_{\text{LLM}}$ takes $(e_{[1:K]}, x, y)$ as input and evaluates the ICL performance for each $e_{[1:K]} \in \mathcal{E}$ through LLM. In this way, we obtain an ICL performance ranking conditioned on $x$. (2) Since the true answer $y$ of a test input $x$ is unavailable during inference, another scoring function

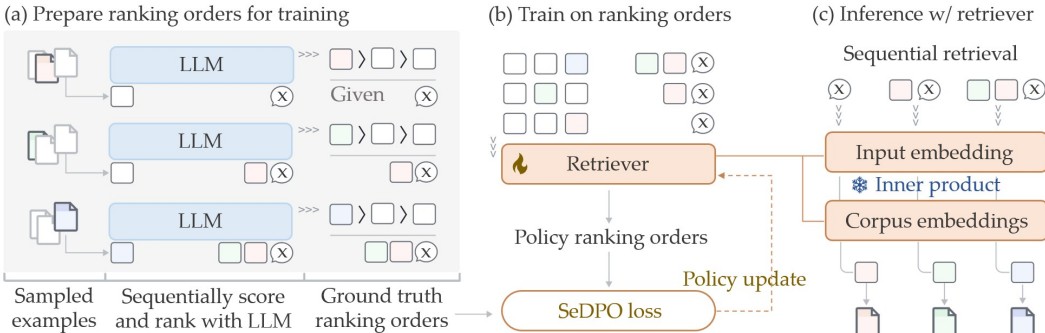

(a) Prepare ranking orders for training  (b) Train on ranking orders  (c) Inference w/ retriever

Figure 2: (a) Prepare preference data $\tilde{\mathcal{D}}$. (b) Train the retriever on $\tilde{\mathcal{D}}$ with SeDPO loss to align with partial order instead of the top-scored. (c) Inference with dense passage retrieval (DPR).

$\phi_{\text{retriever}}$ is introduced, which takes $(e_{[1:K]}, x)$ as input to score each $e_{[1:K]} \in \mathcal{E}$. This produces a retriever-based ranking conditioned on $x$. (3) To ensure that higher-scored retrieved $e_{[1:K]}$ can lead to better ICL performance, our goal is to train the retriever by aligning the $\phi_{\text{retriever}}$-based ranking with the $\phi_{\text{LLM}}$-based ranking. Notably, previous methods widely use classification objective, categorizing $e_{[1:K]}$ with the best ICL performance as positive, and the rest as negative, which is unlike our goal.

## 3.2 Labeling data with ranking orders

**Definition of ranking orders.** We focus on Multiple-Choice Question (MCQ) tasks [44] by default. MCQ is a question format in which respondents (i.e., LLM) are asked to select only the correct answers from the choices offered as a list. How much the LLM's prediction $\hat{y}$ matches the ground truth $y \in \mathcal{Y}_{\text{gt}}$ can be measured by a task-specific scoring function $\text{S}(\cdot, \cdot)$, which can be formulated:

$$\phi_{\text{LLM}}(e_{[1:K]}, x, y) = \text{S}_{\text{MCQ}}(e_{[1:K]} \oplus x, y) = \frac{\text{LH}(y|e_{[1:K]} \oplus x)}{\sum_{y' \in \mathcal{Y}} \text{LH}(y'|e_{[1:K]} \oplus x)}, \quad (2)$$

where $\mathcal{Y}$ or $\mathcal{Y}_{\text{gt}}$ is output/ground-truth label space, LH is per token conditional likelihood of the LLM. For brevity, we denote $e_{[1:K]} \oplus x = (e_K \oplus ... \oplus e_1 \oplus x)$ w.r.t Eq. (1). As higher scores indicate that LLMs are more likely to output ground-truth answers, we can define partial order for ranking as $e_{[1:K]}^w \succ e_{[1:K]}^l | x$ if $\text{S}_{\text{MCQ}}(e_{[1:K]}^w \oplus x, y) > \text{S}_{\text{MCQ}}(e_{[1:K]}^l \oplus x, y)$ for all $y \in \mathcal{Y}_{\text{gt}}$ given $x$. The partial order can be a total order by weighted aggregating $\text{S}_{\text{MCQ}}$ for all $y \in \mathcal{Y}_{\text{gt}}$; but we focus on the partial order as domain knowledge isn't required. In scored MCQ data, there may be multiple or no examples that guide LLMs in predicting correctly, requiring exploration beyond top-scored examples.

**Scored data construction.** Since the corpus $\mathcal{C}$ can form $(|\mathcal{C}|!)/(|\mathcal{C}| - K)!$ possible $e_{[1:K]}$, scoring all $e_{[1:K]}$ would be computationally prohibitive. Following [13], we employ a greedy algorithm to selectively score the data for fair comparison, ultimately obtaining sequentially scored data $\mathcal{D}$. The constructed queries in $\mathcal{D}$ are marked by tilde. For each $(x, y) \in \mathcal{C}$, we denote the test input $x$ without ICE as $\tilde{x}_0$, and sample $L$ examples from corpus $\mathcal{C}$ as $\mathcal{B}(\tilde{x}_0)$. We score the examples in $\mathcal{B}(\tilde{x}_0)$ with frozen LLM and Eq. (2), and repeatedly resample the examples in $\mathcal{B}(\tilde{x}_0)$ that cannot be ranked. Finally, we select an example from scored set $\mathcal{B}(\tilde{x}_0)$ as $e_c$ based on its rank:

$$p(\text{rank}) = \frac{\exp(-\text{rank})}{\sum_{\text{rank}'=1}^{L} \exp(-\text{rank}')} \quad (3)$$

from which the higher-scored example is more likely to be selected and the diversity is preserved. $\tilde{x}_{k+1} = e_c \oplus \tilde{x}_k$ is iteratively updated for next round scoring, until the $K$-shot data are all constructed. The $L$ scored examples in each $\mathcal{B}(\tilde{x}_k)$ are gathered with corresponding $\tilde{x}_k$ to form $\mathcal{D}$ (Fig. 2 (a)).

**Ranked data construction.** Given $\tilde{x}_k$, we have $L$ scored examples, forming $\binom{L}{2}$ pairs of partial order. Empirically, we consider that two types of preference play a dominant role: (a) Given $(x, y)$, examples in different pairs are non-overlapping, so as to enhance the diversity of training data. (b) $e^w$ and $e^l$ have a larger score margin, where their discrepancy is easier to learn. Therefore, we select $T$ examples with the highest scores as the preferred ones and randomly match them with bottom-$T$ dispreferred candidates, to construct sequential preference data $\tilde{\mathcal{D}}$ out of $\mathcal{D}$. We discuss it in ablation.

## 3.3 Algorithm for learning orders

We devise a novel RL algorithm for training a retriever, as directly applying existing RL algorithms such as DPO [12] on ranked data (Sec. 3.2) is prohibitive: (1) DPO requires knowing the probability distribution of specific retrieval actions over the entire corpus (i.e., policy), but the retriever model only outputs retrieval scores for specific ICEs; (2) optimizing retrieval actions over the entire corpus is very expensive and hard to scale. We innovatively address these challenges with (a) reparameterization of retriever score into policy model and (b) sequential relaxation of $e_k \in e_{[1:K]}$ in this section.

**Policy of retriever model.** To train a retriever with ranked data using DPO, we must convert the retriever score into a policy, to model the probability that retrieved ICEs are ranked in the order of ICL performance. We denote $\mathcal{C} \setminus \{e_i\}_{i<k}$ as the corpus excluding the selected ICEs. Based on retrieval scores $\mathrm{sim}(\cdot, \cdot|\cdot)$, the policy $\pi$ of selecting $e_{[1:K]}$ from all candidates without replacement is:

$$\pi(e_{[1:K]}|x) = \prod_{k=1}^{K} \frac{\exp(\mathrm{sim}(e_k, x|\{e_i\}_{i<k}))}{\sum_{e_* \in \mathcal{C} \setminus \{e_i\}_{i<k}} \exp(\mathrm{sim}(e_*, x|\{e_i\}_{i<k}))} \tag{4}$$

**Direct preference optimization** (DPO) [12] is one of the most popular RL methods for preference ranking orders modeling. Instead of learning an explicit reward model, DPO reparameterizes the reward function $r$ using a closed-form expression with the optimal policy. By initializing policy model $\pi_\theta$ and reference model $\pi_{\mathrm{ref}}$ with Eq. (4), the reward function $r$ is as follows

$$r(x, e_{[1:K]}) = \beta \log \frac{\pi_\theta(e_{[1:K]}|x)}{\pi_{\mathrm{ref}}(e_{[1:K]}|x)} + \beta \log Z(x) \tag{5}$$

where $\beta$ controls the KL-divergence constraint on policy/reference models, and $Z(x)$ is the partition function [12]. By incorporating this reward formulation into the BradleyTerry ranking objective, $P(e_{[1:K]}^w > e_{[1:K]}^l|x) = \sigma(r(x, e_{[1:K]}^w) - r(x, e_{[1:K]}^l))$, DPO expresses the probability of preference data with the policy model, yielding the following loss for triple $(x, e_{[1:K]}^w, e_{[1:K]}^l)$:

$$\mathcal{L}_{\mathrm{DPO}}(\pi_\theta; \pi_{\mathrm{ref}}) = -\mathbb{E}_{(x, e_{[1:K]}^w, e_{[1:K]}^l)} \left[ \log \sigma \left( \beta \log \frac{\pi_\theta(e_{[1:K]}^w|x)}{\pi_{\mathrm{ref}}(e_{[1:K]}^w|x)} - \beta \log \frac{\pi_\theta(e_{[1:K]}^l|x)}{\pi_{\mathrm{ref}}(e_{[1:K]}^l|x)} \right), \right] \tag{6}$$

where $e_{[1:K]}^w$ and $e_{[1:K]}^l$ denote the preferred and dispreferred prompts conditioned by input $x$. During training, we update the $\pi_\theta$ while freezing $\pi_{\mathrm{ref}}$. By plugging Eq. (4) into Eq. (6), we have:

$$\mathcal{L}_{\mathrm{DPO}}(\pi_\theta; \pi_{ref}) = -\mathbb{E}_{(x, e_{[1:k]}^w, e_{[1:k]}^l) \sim \mathcal{D}} \left[ \log \sigma \left( \beta \cdot f_\theta(x, e_{[1:K]}^w) - \beta \cdot f_\theta(x, e_{[1:K]}^l) - \beta \cdot (\gamma_w - \gamma_l) \right) \right]$$

$$f_\theta(x, e_{[1:K]}^j) = \sum_{k=1}^{K} \left[ \mathrm{sim}_\theta(e_k^j, x|\{e_i^j\}_{i<k}) - \mathrm{sim}_{\mathrm{ref}}(e_k^j, x|\{e_i^j\}_{i<k}) \right], \qquad j \in \{w, l\}$$

$$\gamma_j = \sum_{k=1}^{K} \log \frac{\sum_{e_* \in \mathcal{C} \setminus \{e_i^j\}_{i<k}} \exp(\mathrm{sim}_\theta(e_*, x|\{e_i^j\}_{i<k}))}{\sum_{e_* \in \mathcal{C} \setminus \{e_i^j\}_{i<k}} \exp(\mathrm{sim}_{\mathrm{ref}}(e_*, x|\{e_i^j\}_{i<k}))}, \qquad j \in \{w, l\} \tag{7}$$

However, the estimation of the policy denominator (i.e., $\gamma$) necessitates repeated re-embedding of the entire corpus for each sampling step. This representation results in a prohibitively high computational overhead. To tackle this, we consider a relaxation by modeling the sequential relation among $e_k$ within the same prompt $e_{[1:K]}$, motivated by left-to-right generation in autoregressive models.

**Sequential LLM preference alignment.** Under the sequential assumption, the ICL performance of the prompt given retrieved $e_k$ is solely influenced by the performance of the retrieved ICEs, i.e., $e_{i<k}$. Therefore, the optimal $e_k \in e_{[1:K]}$ can be explored sequentially:

$$\max_{\{e_k\}_{k=1}^K} \mathrm{S}_{\mathrm{MCQ}}(e_K \oplus ... \oplus e_1 \oplus x, y) \Rightarrow \{e_k | \max_{e_k} \mathrm{S}_{\mathrm{MCQ}}(e_k \oplus ... \oplus e_1 \oplus x, y)\}_{k=1}^K \tag{8}$$

from which sequential preference data $\tilde{\mathcal{D}}$ can be constructed from $\mathcal{D}$ as:

$$\{(e_k^w \succ e_k^l)|e_{k-1} \oplus ... \oplus x\}_{k=2}^K \cup \{(e_1^w \succ e_1^l)|x\} = \{(e_k^w \succ e_k^l)|\tilde{x}_{k-1}\}_{k=1}^K \tag{9}$$

where the conditional term is abbreviated as $\tilde{x}_{k-1}$. Based on this, the policy can be factorized sequentially into $\pi(e_{[1:K]}|x) = \prod_{k=1}^{K} \pi(e_k|\tilde{x}_{k-1})$. Here, $\pi(e_k|\tilde{x}_{k-1})$ is as follows:

$$\pi(e_k|\tilde{x}_{k-1}) = \frac{\exp(\text{sim}(e_k, \tilde{x}_{k-1}))}{\sum_{e_* \in \mathcal{C} \setminus \{e_i\}_{i<k}} \exp(\text{sim}(e_*, \tilde{x}_{k-1}))} \tag{10}$$

by plugging Eq. (10) into Eq. (6), we obtain same form as Eq. (7) with $\gamma_w - \gamma_l = 0$:

$$\mathcal{L}_{\text{SE-DPO}}(\pi_\theta; \pi_{ref}) = -\mathbb{E}_{(\tilde{x}_{k-1}, e_k^w, e_k^l) \sim \tilde{\mathcal{D}}} \left[ \log \sigma \left( \beta \cdot f_\theta(\tilde{x}_{k-1}, e_k^w) - \beta \cdot f_\theta(\tilde{x}_{k-1}, e_k^l) - \beta \cdot 0 \right) \right],$$
$$f_\theta(\tilde{x}_{k-1}, e_k^j) = \text{sim}_\theta(e_k^j, \tilde{x}_{k-1}) - \text{sim}_{\text{ref}}(e_k^j, \tilde{x}_{k-1}), \qquad j \in \{w, l\} \tag{11}$$

The new objective function is tractable by implicitly considering the global partition function (i.e., the denominator in Eq. (10)). Training signals (e.g., $e_k^w, e_k^l$) can be readily generated from scored data. We train the retriever on $\tilde{\mathcal{D}}$ with Eq. (11). The resulting retriever can discern the preferred ICEs for varying context input. Note that the partial order learned by our bi-encoder satisfies the transitivity: if $e^a \succ e^b|x$ and $e^b \succ e^c|x$, then $e^a \succ e^c|x$, ensuring our retriever is ranking-informative. The proof of the transitivity can be found in Appendices.

## 4 Experiments

We take the SOTA method $\text{Se}^2$ [13] as our base, which is a classification-based method, and implement SeDPO on top of it. We first compare the ICL performance with SOTA retrievers (main results); validate key components in SeDPO (ablation); then provide extra experiments for deeper analysis.

### 4.1 Experimental settings

**Task and dataset.** We use a total of 9 tasks across 4 distinct categories, including Paraphrase: **MRPC** [45], **PAWS** [46], **QQP** [47]; Coreference: **WSC** [48]; Reading: **MultiRC** [49], **BoolQ** [50], **AGNews** [51]; NLI: **MNLI-m/mm** [52]. The preprocessing and evaluation for all datasets are the same as $\text{Se}^2$. The description of each dataset is provided in the appendices.

**Implementation details.** For fair comparison, we follow the hyperparameter setting of $\text{Se}^2$ and use GPT-Neo-2.7B [53] as the scoring and inference LLM for most of the experiments. The encoders in retriever model are initialized with "BERT-base-uncased" [43]. The shot number $K = 3$. We set $T = 20$, ensuring that $\tilde{\mathcal{D}}$ is sourced from the same training data for fairness; retriever is fine-tuned for 6 epochs on each category, the best checkpoint is chosen based on retrieval accuracy of validation set, and evaluated using task-specific metric on the test set. Refer to appendices for more details.

Additionally, $\text{Se}^2$ incorporates two data augmentation techniques: (1) positive chosen — $\text{Se}^2$ reserves only the data where the selected representative examples can guide LLM in correctly predicting; (2) in-batch rejection — in each training batch, the negative sample set for each input is extended with examples from other inputs for diversity. Following $\text{Se}^2$, we reserve the data where $e^w$ guides LLM in correctly predicting; for each input, the rejected instance is uniformly sampled from combined set of original $e^l$ and examples from other inputs. We analyse the techniques in ablation study.

**Baselines.** All model comparisons are fair using the same size, as detailed in the appendices. We repeat each algorithm 10 times and report the average performance. The off-the-shelf baselines include: **Random** involves sampling $K$ demonstrations randomly; **BM25** [21] is the commonly used sparse retriever that finds exemplars based on textual similarity; **SBERT** [23] is a dense retriever by computing sentence embedding, we take "paraphrase-mpnet-base-v2" as its encoder.

For fair comparison, we re-trained the learning-based baselines aligning our task settings (e.g., with the same scored dataset): **UPRISE** [29] estimate the usefulness of each example separately; **$\text{Se}^2$** [13] improves upon UPRISE by sequentially retrieving representative examples, making it the most competitive and accessible alternative for comparison; **UDR** [35] uses LambdaRank loss to inject ranking information, while estimating the usefulness of each example separately. We also study the integration of UDR's LambdaRank loss with $\text{Se}^2$ in the ablation experiments.

Table 1: Main results on various tasks. The **best results** and the second-best are highlighted. The *Avg.* of all metrics for tasks within the same category with significant improvements is marked by ↑.

| | Paraphrase | | | | | | Coreference |
|---|---|---|---|---|---|---|---|
| | **MRPC** | | **PAWS** | **QQP** | | *Avg.* | **WSC** |
| | acc | f1 | acc | acc | f1 | | acc/*Avg.* |
| Zeroshot | 46.1±0.0 | 45.3±0.0 | 51.8±0.0 | 48.4±0.0 | 42.1±0.0 | 46.7±0.0 | 59.6±0.0 |
| Random | 66.8±3.0 | 79.5±4.1 | 50.1±3.8 | 40.6±4.8 | 50.9±7.8 | 57.6±3.8 | 48.3±8.2 |
| BM25 | 57.8±0.0 | 69.1±0.0 | 48.9±0.0 | 54.8±0.0 | 55.4±0.0 | 57.2±0.0 | 52.4±0.0 |
| SBERT | 56.4±0.0 | 66.9±0.0 | 49.4±0.0 | 51.2±0.0 | 56.2±0.0 | 56.0±0.0 | 46.2±0.0 |
| UDR | 65.9±4.6 | 75.4±3.5 | 51.8±1.2 | 74.1±1.9 | 67.9±2.4 | 67.0±1.2 | 52.0±4.7 |
| UPRISE | 74.0±0.8 | 83.3±0.1 | 49.1±0.0 | 71.0±1.0 | 69.8±0.1 | 69.4±0.2 | 46.5±2.2 |
| Se$^2$ | 77.6±0.4 | 85.4±0.3 | 54.7±0.1 | 75.5±0.1 | 72.8±0.0 | 73.2±0.2 | 55.1±0.9 |
| SeDPO | **77.9±0.9** | **85.6±0.2** | **73.0±2.9** | **77.6±0.6** | **75.0±0.2** | **77.9±0.6**[↑] | **62.5±0.2**[↑] |

| | Reading | | | | Natural Language Inference (NLI) | | |
|---|---|---|---|---|---|---|---|
| | **MultiRC** | **BoolQ** | **AGNews** | *Avg.* | **MNLI-m** | **MNLI-mm** | *Avg.* |
| | f1 | acc | acc | | acc | acc | |
| Zeroshot | 57.1±0.0 | 54.6±0.0 | 38.4±0.0 | 50.0±0.0 | 35.2±0.0 | 36.4±0.0 | 35.8±0.0 |
| Random | 57.7±2.5 | 54.8±6.7 | 25.8±1.1 | 46.1±1.2 | 34.2±3.0 | 34.9±3.9 | 34.6±1.6 |
| BM25 | 46.5±0.0 | 60.3±0.0 | 81.7±0.0 | 62.8±0.0 | 35.3±0.0 | 35.6±0.0 | 35.5±0.0 |
| SBERT | 49.3±0.0 | 58.1±0.0 | 84.7±0.0 | 64.0±0.0 | 37.3±0.0 | 37.3±0.0 | 37.3±0.0 |
| UDR | 55.3±3.1 | 54.6±1.9 | 88.5±1.0 | 66.1±0.9 | 62.7±1.5 | 65.0±1.3 | 63.8±1.4 |
| UPRISE | 55.4±0.2 | 61.5±0.1 | 90.6±0.8 | 69.2±0.1 | 68.5±0.1 | 70.3±0.3 | 69.4±0.2 |
| Se$^2$ | 52.1±2.3 | 63.6±0.2 | 90.8±0.3 | 68.8±0.7 | 69.4±0.2 | 70.4±0.1 | 69.9±0.2 |
| SeDPO | **60.3±0.4** | **64.6±1.7** | **91.0±0.2** | **72.0±0.6**[↑] | **70.6±0.1** | **72.0±0.3** | **71.3±0.2**[↑] |

## 4.2 Main results

Table 1 details the experimental results on MCQ tasks, where generative LLMs are known to need improvement [29]. On each task, we mark the best results in bold and underline the second-best. The *Avg.* column represents the mean performance for each category, with significant improvements marked by [↑] (confidence level is 99%) over the best alternative. SeDPO outperforms all other methods in all categories with an avg improvement of up to 4.7%, and we also have several findings.

First, random sampling does not lead to sizable gains compared to zeroshot. In contrast, BM25 and SBERT have a significant gain over random sampling and zeroshot. This demonstrates the necessity of providing related examples for LLMs in downstream tasks. In addition, finetuning-based retrievers perform better than off-the-shelf retrievers, highlighting the effectiveness of using LLM feedback.

Second, among the finetuning-based retrievers, Se$^2$ outperforms UPRISE by using sequential relationships between examples, showing the importance of capturing sequential relationships between examples. Though UDR considers local ranking regularization, the performance gain is limited on average compared to Se$^2$. In particular, our proposed SeDPO achieves the best performance in all categories, significantly exceeding Se$^2$ by at least 1.4% to a maximum of 7.4%. This demonstrates that learning preference orders rather than the representative examples, significantly enhances the ICL performance on MCQ tasks and complements previous advancements.

Third, for downstream tasks such as PAWS, WSC, and MultiRC, SeDPO achieves a substantial improvement of up to 18.3%. We speculate that retrievers trained to learn representative patterns fail to capture the true preferences of LLMs on challenging data, as they show no significant improvement compared to zero-shot. Particularly, Se$^2$ underperforms random selection in both MultiRC and WSC tasks, indicating that these tasks require the capture of more diverse query-dependent ICE patterns. Se$^2$ learns from top-scored ICEs, thus insufficient to generalize to test input. In contrast, SeDPO consistently outperforms random selection by 2.9% to 3.9% and surpasses Se$^2$ by 7.4% to 8.2%. This demonstrates the superiority of our proposal in challenging tasks.

## 4.3 Ablation studies

We compare our original framework with its variants altering each time a different component.

Table 2: Ablation results on *Paraphrase* task using GPT-Neo-2.7B, trained on 3-shot.

|  | *Paraphrase* |
|---|---|
| SeDPO ($\beta = 0.02$) | 77.9 |
| w/o positive chosen | 77.9 |
| w/ top-1 chosen | 70.8 |
| w/o in-batch rejection | 73.9 |
| w/ random preference | 57.6 |
| w/ LambdaRank (UDR) | 72.9 |
| w/ RoBERTa | 85.7 |
| SeDPO ($\beta = 0.02$) $\circ$ Se$^2$ | 79.0 |
| Se$^2$ $\circ$ SeDPO ($\beta = 0.02$) | 74.8 |

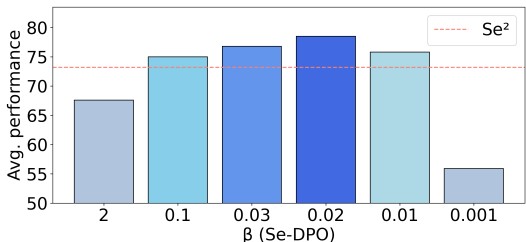

Figure 3: Performance of SeDPO on *Paraphrase* category using GPT-Neo-2.7B with different $\beta$. The dashed line represents the results of **Se$^2$**.

Table 3: The average textual/semantic diversity of selected ICEs, as well as the average performance when the input order of ICEs is randomized. We take the main results on *Paraphrase* as our base.

|  | **SeDPO** | **Se$^2$** | **UPRISE** | **UDR** | **SBERT** | **BM25** | **Random** |
|---|---|---|---|---|---|---|---|
| Textual Diversity | 53.3% | 49.0% | 46.7% | 54.3% | 49.7% | 46.0% | **61.4%** |
| Semantic Diversity | 40.7% | 39.0% | 37.3% | 40.3% | 25.3% | 29.0% | **46.0%** |
| Random order (Best of 5) | 78.2% | 73.5% | 70.9% | 68.4% | 57.3% | 58.1% | - |
| Random order (Worst of 5) | 77.1% | 72.5% | 68.6% | 66.3% | 55.1% | 57.0% | - |

**Complementary strength.** We test the complementary strength of SeDPO and Se$^2$. In Table 2, SeDPO ($\beta$=0.02) $\circ$ Se$^2$ denotes initializing the retriever's weights of Se$^2$ with weights of trained SeDPO, and vice versa. Training Se$^2$ model using SeDPO's trained weights can enhance ICL performance. We surmise that the greedy data construction may not be globally optimal for fully learning preference ranking order, there is still room for improvement. Conversely, initializing SeDPO with Se$^2$'s weights leads to suboptimal results. This asymmetry likely arises from SeDPO's inherent KL-divergence constraint, which preserves the retriever's pretrained knowledge base. Se$^2$ may overfit to top-scored, hindering SeDPO's ability to use the retriever's pretrained knowledge.

**Effect of components.** As shown in Table 2, replacing our original preference data with randomly selected pairs of the same size led to poor results (*w/ random preference*). Since $T \ll \binom{L}{2}$, random local observations struggle to learn reliable partial orderings. Furthermore, using local representative examples (*w/ top-1 chosen*) as the only chosen in preference data and pairing them randomly with bottom-$T$ examples, while performing better than random selection, still falls short of our original design. This demonstrates the effectiveness of reserving diverse $(e^w, e^l)$ with a larger score margin.

Notably, we also investigate the impact of the two data augmentation techniques used in Se$^2$ (detailed in Sec. 4.1). The *positive chosen* does not enhance our results, indicating that learning orders rely more on the discrepancy between chosen/rejected samples rather than the absolute quality of the chosen ones. In contrast, *in-batch rejection* increases data diversity and improves our results by 4%.

**Influence of embeddings.** This paper uses "BERT-base-uncased" as the encoder model for fair comparison. Nowadays there are many better text embedding models available. To further show the influence of the embeddings, we replaced the retriever backbone of SeDPO with "RoBERTa-base" [54], which is known to outperform "BERT-base-uncased" [43] across a range of NLP tasks. The results are shown as *w/ RoBERTa* in Table 2, SeDPO benefits from stronger embeddings as expected.

**Diversity of retrieved examples.** Table 3 presents the diversity metrics corresponding to the main results in Table 1, where higher values indicate lower query-similarity and a richer context of retrieved ICEs. Both SeDPO and UDR achieve high diversity in selected ICEs, exceeding other retrievers (except random) by at least 3.6%/1.3% in textual/semantic diversity. This demonstrates that incorporating ranking signals enhances diversity. The poor ICL performance of randomly selected ICEs highlights the importance of selecting relevant ICEs. In addition, though UDR's diversity is comparable to SeDPO's, its ICL performance lags behind Se$^2$ and UPRISE. This indicates that SeDPO better trades off diversity with ICL utility and successfully retrieves *diverse yet useful* ICEs.

Table 4: Transferability on shot number and model size. The average performance of *Paraphrase*.

| Inference Model | Method | 1-shot | 3-shot | 6-shot | 9-shot | 12-shot | 15-shot | Average |
|---|---|---|---|---|---|---|---|---|
| GPT-2-XL-1.5B (0-shot=39.6) | BM25 | 57.6 | 58.5 | 58.8 | 58.7 | 59.3 | 60.1 | 58.8 |
| | SBERT | 57.9 | 57.5 | 59.0 | 59.6 | 58.6 | 58.3 | 58.5 |
| | UPRISE | 69.2 | 69.4 | 69 .8 | 69.8 | 70.0 | 70.2 | 69.7 |
| | $Se^2$ | 73.9 | 72.9 | 72.9 | 72.8 | 72.8 | 72.7 | 73.0 |
| | SeDPO | 75.0 | 78.9 | 79.5 | 79.2 | 79.0 | 79.2 | **78.5** |
| GPT-Neo-2.7B (0-shot=46.7) | BM25 | 57.1 | 57.2 | 58.9 | 59.5 | 59.0 | 59.4 | 58.5 |
| | SBERT | 56.6 | 56.0 | 59.4 | 58.9 | 59.8 | 58.4 | 58.2 |
| | UPRISE | 69.4 | 69.7 | 69.5 | 69.2 | 69.2 | 69.3 | 69.4 |
| | $Se^2$ | 73.5 | 73.2 | 73.1 | 73.0 | 72.8 | 72.6 | 73.0 |
| | SeDPO | 77.6 | 77.9 | 78.0 | 77.9 | 78.2 | 78.1 | **78.0** |
| Llama3-8B-Instruct (0-shot=56.4) | BM25 | 68.6 | 73.2 | 74.7 | 75.1 | 75.6 | 76.7 | 74.0 |
| | SBERT | 68.3 | 73.0 | 73.4 | 75.1 | 75.4 | 76.1 | 73.5 |
| | UPRISE | 70.9 | 75.3 | 76.4 | 76.6 | 76.9 | 77.0 | 75.5 |
| | $Se^2$ | 71.9 | 76.7 | 78.0 | 78.0 | 77.9 | 77.9 | 76.7 |
| | SeDPO | 71.9 | 77.4 | 78.5 | 79.3 | 80.2 | 80.3 | **77.9** |
| Llama3.3-70B (0-shot=67.6) | BM25 | 78.4 | 80.7 | 82.2 | 81.7 | 81.8 | 82.2 | 81.2 |
| | SBERT | 78.7 | 80.3 | 81.2 | 81.7 | 81.7 | 82.7 | 81.1 |
| | UPRISE | 77.3 | 80.2 | 81.3 | 80.5 | 80.8 | 81.3 | 80.2 |
| | $Se^2$ | 77.9 | 81.0 | 82.0 | 82.2 | 81.9 | 81.9 | 81.1 |
| | SeDPO | 78.6 | 81.0 | 82.3 | 82.9 | 83.2 | 83.2 | **81.9** |

**Impact of ICE ordering.** We empirically analyze the impact of randomizing the input order of ICE on paraphrase performance. The results in Table 3 suggest that randomizing the ICE order exhibits limited potential for enhancing sequential approaches. We attribute this to that SeDPO and $Se^2$ already optimize the input order of ICEs to an extent by sequential example selection.

**Impact of $\beta$.** Table 3 shows the *Paraphrase* performance of SeDPO using GPT-Neo-2.7B with different $\beta$. Too large or too small $\beta$ can lead to a dominant or negligible constraint of KL divergence, resulting in performance degradation. SeDPO shows promising improvements when $\beta$ ranges from 0.01 to 0.1, so we tune $\beta$ between 0.001 and 2 across different categories.

**Impact of $T$.** Table 12 in **Appendices** shows the extra ablation results for different $T$, following the setup of Table 2. As $T$ increases, the performance of SEDPO improves because a broader preference ranking is learned. However, when $T$ is too large, the performance gain decreases due to more low-confidence LLM rankings, leading to the same conclusion as in Table 2 (*w/ random preference*).

## 4.4 Analysis

**Transferability.** As LLM scales, aligning the LLM preference under different shot numbers $K$ of ICEs is time-consuming and resource-intensive. We thus explore the effectiveness of retrievers, as LLMs and example numbers vary. Specifically, in our main experiments (Table 1), we trained retrievers for each category using 3-shot data and GPT-Neo-2.7B. We then evaluated these retrievers in unseen inference settings, where the $K$ varied from 1 to 15, and the inference LLMs included GPT-2-XL [55], GPT-Neo-2.7B, Llama3-8B-Instruct and Llama3.3-70B [2]. Table 4 illustrates that SeDPO consistently outperforms baseline retrievers in all settings. Notably, though SeDPO is trained on 3-shot, its performance improves as the $K$ increases and significantly outperforms 0-shot by at least 10% on all LLMs. We also find that as the model size of LLMs increases, the gap between different ICL methods decreases. This indicates that larger LLMs are smarter and can reason using suboptimal examples; on the other hand, smaller LLMs (not that smart) rely more on high-quality examples, as is also mentioned in [56]. However, the in-context examples provided by SeDPO are useful for both small and large LLMs, demonstrating strong transferability.

**Math inference.** We compare SeDPO and $Se^2$ with several reasoning-focused retrievers [57, 58, 28] on AUQA [59], a MCQ task requiring math inference. Table 9 in **Appendices** shows that SeDPO outperforms $Se^2$, aligning with our main discovery. Though RGER [57] is built for reasoning-focused tasks, SeDPO surpasses RGER [57] without being geared toward reasoning capabilities.

**More in-depth analysis.** To further study the performance boundaries of SeDPO, we provide extra results in **Appendices**, including 0-shot performance on human-labeled data, impact of preference

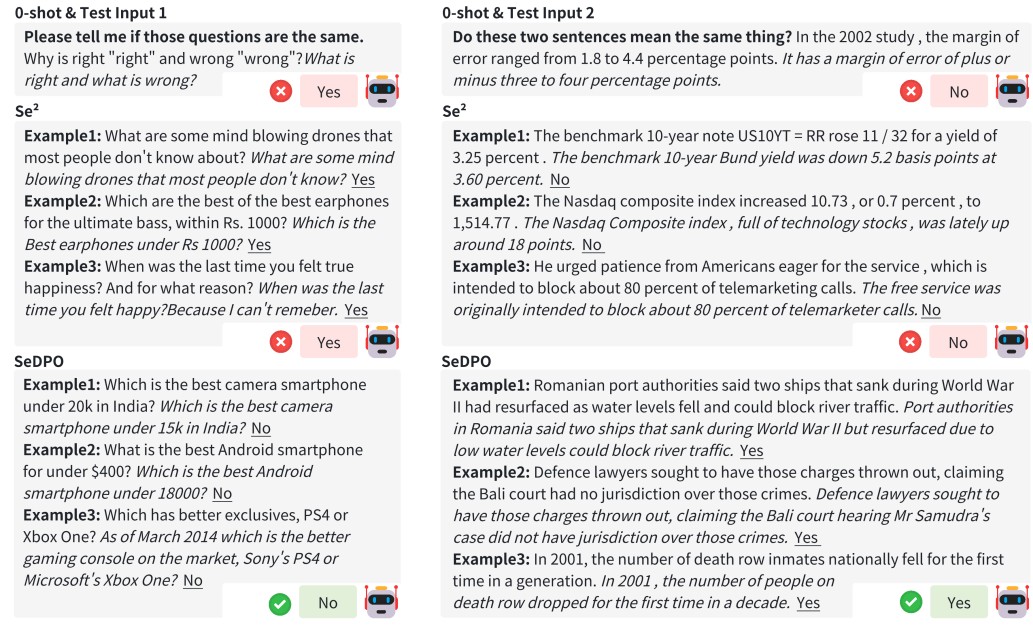

Figure 4: Two cases on *Paraphrase* where SeDPO helps LLM correctly infer, but Se$^2$ does not.

dataset construction using various LLMs, detailed math inference, analysis of full list order, proof of transitivity, cost of constructing training data, discussion of Eq. (2) in open-ended QA setting.

**Case study.** In Figure 4, we analyze two cases from the *Paraphrase*, to intuitively compare the effectiveness of SeDPO and Se$^2$. The answers of the examples are marked with underscores. Specifically, the task focuses on whether two sentences in the test input are synonymous. Se$^2$'s examples emphasize surface patterns: in the left case, the two sentences have similar word compositions; in the right case, there are changes in percentage numbers. In contrast, SeDPO captures the task-related nuances preferred by LLMs: it selects examples based on the extent of difference between the two sentences and marginalizes causal information using broader contexts such as historical, legal, and social. This demonstrates how learning preference ranking orders gives a broader causal relationship between examples, which improves the ICL performance. More cases can be found in the appendices.

## 5 Conclusion

We considered learning to rank for in-context example retrieval, introducing SeDPO, a simple yet effective method. Unlike dominant methods that focus on representative examples, SeDPO captures the global preference orders through a pairwise ranking formulation. We effectively address the issue that classification-based retrievers poorly capture broader utility. Extensive experiments demonstrate our superiority. **Additional experiments, discussions, and proofs** appear in the Appendices.

**Limitations.** Our research focuses on improving retriever training but shares existing frameworks' structural limitations. First, we mainly use GPT-Neo-2.7B for fair comparison, where shot number analysis is constrained by sequence length; this can be improved by recent input/prompt compression [60]. Second, results are affected by inherent biases [61] in retriever models and LLMs, requiring fair and interpretable strategies (a promising direction). Third, permutation-based example retrieval is underexplored. Notably, SeDPO needs dominant preference identification to reduce computational costs and redundant interference; while clear for MCQ tasks, open-ended question challenges remain.

**Broader Impacts.** Learning-to-rank for ICE retrieval boosts LLM's ICL performance but carries negative societal risks. Without safeguards, it may retrieve/prioritize biased, misleading, or harmful examples—reinforcing unfair decisions (e.g., employment/legal consultation) or spreading disinformation—and could be misused to get adversarial examples manipulating LLMs. Mitigation requires rigorous fairness audits, retrieved example filters, and controlled access for high-stakes deployments.

## Acknowledgments and Disclosure of Funding

This work was supported by the National Natural Science Foundation of China (NSFC) Key Program under Grant Number 62336006, the National Key R&D Program of China (2021YFB3500700), and the Beijing Science and Technology Plan Project.

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

# Appendices

## .1 Datasets details

- **MRPC**: A paraphrase task with 3,668 training examples and 408 test examples, evaluated using Accuracy and F1.
- **PAWS**: A paraphrase task with 49,401 training examples and 8,000 test examples, evaluated using Accuracy.
- **QQP**: A paraphrase task with 363,846 training examples and 40,430 test examples, evaluated using Accuracy and F1.
- **WSC**: A coreference task with 554 training examples and 104 test examples, evaluated using Accuracy.
- **MultiRC**: A reading comprehension task with 27,243 training examples and 4,848 test examples, evaluated using F1.
- **BoolQ**: A reading comprehension task with 9,427 training examples and 3,270 test examples, evaluated using Accuracy.
- **AGNews**: A reading comprehension task with 120,000 training examples and 7,600 test examples, evaluated using Accuracy.
- **MNLI-m/mm**: A natural language inference task with 392,702 training examples and 9,815/9,832 test examples for m/mm, evaluated using Accuracy.

We list the detailed datasets' statistical information above. To convert datasets into natural language instructions, we follow previous practice [13] using the instruction template of FLAN [62]. Each task dataset corresponds to approximately seven templates. All datasets are publicly available under open licenses (e.g., CC-BY, CC-BY-SA, or research-only terms). Datasets are all for MCQ tasks and are widely used in relevant work without offensive content, which is in line with our purpose of use.

## .2 Model details

- **BERT-base-uncased** has approximately 110 million parameters and is released under the Apache License 2.0.
- **GPT-Neo-2.7B** consists of 2.7 billion parameters and is distributed under the MIT License.
- **Llama3-8B-Instruct** features 8 billion parameters and is licensed under the Meta Llama 3 Community License.
- **Llama3.3-70B** contains 70 billion parameters and is also governed by the Meta Llama 3 Community License.

The documentation for the artifacts is publicly available, refer to their citations in main paper.

## .3 Implementation details

Table 5 lists the overall hyperparameters. For fair comparison, we follow the hyperparameter setting of $Se^2$: we use GPT-Neo-2.7B [53] as the scoring and inference LLM for most experiments; both encoders were initialized with "BERT-base-uncased" [43]; up to 10k data points are randomly selected for each task to construct the training data and example pool for each category while maintaining class balance in classification tasks; sample size $L$ is set to 50 by default, depending on the configuration of $Se^2$; the shot number $K=3$; retriever is fine-tuned for 6 epochs on each category, the best checkpoint is chosen based on retrieval accuracy using the validation set, and evaluated using task-specific metric on the test set. For the hyperparameters of SeDPO, we set $T=20$ and ensure that our preference data are sourced from the training data of $Se^2$ for data fairness. $\beta$ takes values between 0.001 and 2.

**Number of samples per batch.** Note that in mini-batch training, for each top-scored positive example considered by $Se^2$, it is necessary to simultaneously consider $T$ low-scored negative examples and $T$ negative examples randomly sampled from the corpus, as shown in parentheses of # of samples per batch for $Se^2$ in Table 5. SeDPO generates $T$ positive and negative example pairs for each input, but each positive example only needs to consider one negative example, as shown in parentheses of the number of samples per batch for SeDPO in Table 5. We trained the retriever with 8 threads in a

Table 5: Hyperparameter settings.

| Hyperparameter | Assignment | Hyperparameter | Assignment |
|---|---|---|---|
| shot-number | 3 | Max sequence length | 512 for retriever |
| Optimizer | Adam | | 2048 for LLMs |
| Number of epochs | 6 | per GPU Max learning rate | 1e-5 |
| Preference $\beta$ | $> 0.001$ | # of samples per batch for SeDPO | 8*32*(1+1) |
| | $< 2$ | # of samples per batch for Se$^2$ | 1*32*(1+2*$T$) |
| Adam epsilon | 1e-8 | Warmup steps | 1000 |
| Adam beta weights | 0.9, 0.999 | Learning rate decay | linear |
| Weight decay | 0.0 | Learning rate scheduler | warmup linear |

data-distributed manner on 8*A100-80GB. To speed up the training process of Se$^2$, we considered 32 positive examples in each mini-batch, along with their dependent negative examples, resulting in a total of 32*(1+2*$T$) examples used per mini-batch. Since SeDPO needs to allocate additional memory to the reference model, we only consider using 8*32*(1+1) examples per mini-batch in SeDPO. The number of samples per batch can be adjusted according to the experimental environment.

Since the training algorithm does not alter the model architecture, the total number of parameters remains 220M, consistent with Se$^2$. Briefly put, our design is compatible with existing fine-tuning-based retrievers using DPR[27] or its sequential version, no further inference or data load is introduced. SeDPO leaves the original framework's token flux unchanged [13], each task takes about 7/9 hours in the scoring/training phase. To allay concerns that the improved ICL performance might stem from differences in backbone models, we detail settings of related methods in Table 6 for reference only.

Table 6: The setting of the related methods.

| | Finetuned | Retriever (Number of Parameters) | Scoring LLM |
|---|---|---|---|
| BM25 | ✗ | ✗ | ✗ |
| SBERT | ✗ | paraphrase-mpnet-base-v2 (1*109) | ✗ |
| UPRISE | ✓ | 2*BERT-based-uncased (2*110M) | GPT-Neo-2.7B |
| Se$^2$ | ✓ | 2*BERT-based-uncased (2*110M) | GPT-Neo-2.7B |
| SeDPO | ✓ | 2*BERT-based-uncased (2*110M) | GPT-Neo-2.7B |

## .4 Diversity calculation

Textual diversity in Table 3 is measured by Levenshtein Edit distance:

$$div_{\text{textual}}(s_1, s_2) = \frac{dist(s_1, s_2)}{sum(len(s_1), len(s_2))} \tag{12}$$

where $s_1$ and $s_2$ are two sentences, $dist(\cdot, \cdot)$ is Levenshtein Edit distance, $len(\cdot)$ denotes the number of characters in sentence. Semantic diversity in Table 3 is measured by SBERT [23]:

$$div_{\text{semantic}}(s_1, s_2) = \frac{1 - cos(E(s_1), E(s_2))}{2} \tag{13}$$

where $E(\cdot)$ is the sentence embedding encoded by SBERT and $cos(\cdot, \cdot)$ is the cosine similarity between two embeddings. We take "paraphrase-MiniLM-L6-v2" as the encoder.

Table 7: The co-reference performance using preference data generated by different LLMs.

| | GPT-Neo-2.7B | Llama3-8B-Instruct | DeepSeek-R1 |
|---|---|---|---|
| **SeDPO** | 62.5% | 59.6% | 53.8% |
| **Se$^2$** | 55.1% | 54.8% | 53.8% |

## .5 Analysis of preference labeling

It will be valuable to study the reliability of the preferences generated by artificial intelligence and their impact on performance. To this end, we conduct comparative experiments on dataset

construction using various LLMs in two settings: (a) To study the reliability of the likelihood-based scoring method, we construct training data through Llama3-8B-Instruct [2] using Eq. (2). (b) To compare with the prompt-engineer-based scoring method, we use DeepSeek-R1 as an agent to simulate a human annotation pipeline following Table 8. The overall results on the co-reference task are shown in Table 7. Specifically, to align with the results in Table 1, the test time LLM is still GPT-Neo-2.7B. The generalization ability of the retriever on different test LLMs has been analyzed in Table 4. The results show that imitating human sorting cannot super-enhance ICL performance. We recommend using the output probability of the LLM for the desired answer to rank ICEs.

Table 8: Prompt used for R1 ranking.

| Prompt |
| --- |
| You have a question inside <question> tags, and you have a correct answer inside <answer> tags. Your task is to determine which examples inside <demonstrations> tags are more conducive to obtaining the correct answer to the question. |
| <question>{#question}<\question> <answer>{#answer}<\answer> <demonstrations>#demonstrations<\demonstrations> |
| In the tags, the examples are arranged in the form of "ID: example". You can sort all the examples based on their usefulness and return their sorted IDs in the form of a number list that can be parsed by JSON in the <output> tags. More useful examples should be at the front of the list. |

**Discussion on full list order.** Recent advances [63] in Reinforcement Learning from Human Feedback show the potential of modeling full list order. The policy in this kind of method requires estimating the partition function by knowing all actions. For LLMs, the partition function means being able to observe the entire vocabulary to model the probability of the next word, and this is achievable. However, applying such a formulation to retrieval is substantially more challenging — it involves re-embedding the entire corpus at each sampling step, which leads to prohibitive computational costs. In contrast, SeDPO, the pairwise approach, circumvents this by implicitly estimating the partition function, i.e., the denominator of retrieval policy in Eq. (4). This makes pairwise algorithm feasible.

Table 9: The performance on mathematical inference tasks.

| | Base LLM | Base Shot Number | AUQA-3shot | AUQA-8shot |
| --- | --- | --- | --- | --- |
| CEIL [28] | Llama2-7B-chat | 8 | - | 22.83 |
| DQ-LoRe [58] | Llama2-7B-chat | 8 | - | 25.20 |
| RGER [57] | Llama2-7B-chat | 8 | - | 25.59 |
| Se$^2$ | GPT-Neo-2.7B | 3 | 24.80 | 22.44 |
| SeDPO | GPT-Neo-2.7B | 3 | 25.59 | 27.56 |

## .6 Mathematical inference

The results on inference tasks are shown in Table 9. Due to variations in the language models, tasks, instruction templates, training and testing datasets, as well as evaluation metrics used by different methods, and due to limitations in computational resources, it is hard to include all related work in the comparison. So we also collected the settings and reported performance of [57] for reference only. Note that only the AUQA [59] task adopted in RGER [57] belongs to MCQ and is compatible with our scoring framework. The results show SeDPO still outperforms Se$^2$ on inference tasks, aligning with our main discovery. Notably, though RGER is explicitly built for reasoning-focused tasks, SeDPO achieves comparable effectiveness without being geared toward reasoning capabilities.

## .7 0-shot retrieval performance

BEIR [64] is a robust and heterogeneous evaluation benchmark for information retrieval, aiming to assess the 0-shot retrieval capabilities on human-labeled document of retrieval models. We

Table 10: The zero-shot retrieval performance on tasks of BEIR.

| | webis-touche2020 | fiqa | scidocs | arguana | nq | Avg. |
|---|---|---|---|---|---|---|
| DPR-BERT | 0.0000 | 0.0002 | 0.0025 | 0.0601 | 0.0012 | 0.0128 |
| Se$^2$ | 0.0000 | 0.0000 | 0.0032 | 0.0287 | 0.0010 | 0.0066 |
| SeDPO | 0.0000 | 0.0000 | 0.0032 | 0.0701 | 0.0013 | 0.0149 |

evaluate nDCG@10 following BEIR in Table 10. While both Se$^2$ and SeDPO are fine-tuned on DPR-BERT, they show no significant 0-shot retrieval gains, aligning with BEIR's observation about dense retrievers' generalization limitations on human-labeled documents. Furthermore, as Table 3 analyzed, SeDPO and Se$^2$'s task-preference specialization can compromise semantic similarity modeling, which may be detrimental to content-similarity tasks in BEIR. Addressing this requires strategies such as scaling and task-aware prompting, which presents a promising research direction.

## .8 Extra analyse

**Discussion of $\phi_{LLM}$ in open-ended QA setting.** Our paper focused on MCQ setting, with theoretical soundness. We also empirically analyze open-ended QA as follows. For open-ended QA, the set $\mathcal{Y}$ becomes impractically large, making Eq. (2) intractable to compute. We have experimented with sampling $y$ values for $\mathcal{Y}$ directly from the model; however, the results were unsatisfactory. Unlike MCQs, open-ended QA lacks a clear way to quantify and normalize the quality gap between good and bad answers, making reliable supervision difficult.

**Proof of transitivity.** By the definition of $\succ$ (Section 3.2):

- If $e^a \succ e^b \mid x$, then for all $y \in \mathcal{Y}_{\text{gt}}$, $S_{\text{MCQ}}(e^a \oplus x, y) > S_{\text{MCQ}}(e^b \oplus x, y)$. (1)
- If $e^b \succ e^c \mid x$, then for all $y \in \mathcal{Y}_{\text{gt}}$, $S_{\text{MCQ}}(e^b \oplus x, y) > S_{\text{MCQ}}(e^c \oplus x, y)$. (2)

Note that $S_{\text{MCQ}}(\cdot)$ is a scalar-valued function, and its outputs are real numbers. The ">" relation on the real numbers is transitive: for any real numbers $a, b, c$, if $a > b$ and $b > c$, then $a > c$.

Applying this transitivity to (1) and (2) for each $y \in \mathcal{Y}_{\text{gt}}$:

For all $y \in \mathcal{Y}_{\text{gt}}$, $S_{\text{MCQ}}(e^a \oplus x, y) > S_{\text{MCQ}}(e^b \oplus x, y)$ and $S_{\text{MCQ}}(e^b \oplus x, y) > S_{\text{MCQ}}(e^c \oplus x, y)$ implies $S_{\text{MCQ}}(e^a \oplus x, y) > S_{\text{MCQ}}(e^c \oplus x, y)$. (3)

By the definition of $\succ$ again, (3) implies $e^a \succ e^c \mid x$. Thus, the relation $\succ$ is transitive.

Q.E.D.

**Cost of constructing training data.** We provide in Table 11 the average cost of constructing preference data for all tasks. The scoring batch size is 10, using GPT-Neo-2.7B as the ICL model. For each $x$, we only sample $T = 20$ preference pairs. This means the number of processed preference pairs is less than the number of scored entries. For instance, on NLI, the total time for constructing scored data is 794× greater than that for preference data:

Table 11: Cost of constructing training data.

| Steps to construct trainset | Speed |
|---|---|
| Score data (Se2) | 0.0241 s/entry |
| Rank scored data (SeDPO) | 0.0007 s/pair |

Table 12: Impact of $T$

| Method | T=1 | T=10 | T=20 | T=30 |
|---|---|---|---|---|
| SeDPO ($\beta$=0.02) | 66.1 | 74.5 | 77.9 | 79.9 |

## .9 More case studies

In addition, we provide more interesting cases in Table 13. Various finetuning-based baselines of our experiments are considered.

Table 13: More case studies on different tasks.

| **Task: Paraphrase (MRPC)** |
|---|

Test Input: "And it's going to be a wild ride," said Allan Hoffenblum, a Republican consultant. "Now the rest is just mechanical," said Allan Hoffenblum, a Republican consultant. Please tell me if the sentences above mean the same.

**UPRISE**:
Example1: His wife, who he married in a first ever space wedding by a space phone during his lengthy mission, waited in Moscow. His wife Yekaterina Dmitriyeva, whom he married in a first ever space wedding by a space phone during his daunting mission, was waiting for him in Moscow. Please tell me if the sentences above mean the same. Yes.

Example2: The main change, said Jim Walton, CNN's president, is a fundamental shift in the way CNN collects its news. The main change, said CNN president Jim Walton, was a fundamental shift in the way the network collected its news. Please tell me if the sentences above mean the same. Yes.

Example3: "There were," said board member and Nobel-prize winning Stanford physicist Douglas Osheroff, "some extremely bad decisions." Board member Douglas Osheroff, a Nobel-prize winning Stanford physicist, said: "There were some extremely bad decisions." Please tell me if the sentences above mean the same. Yes.

Test Input Answer: Yes. ✗

**Se²**:
Example1: The Nets and the Spurs are crossing new frontiers of offensive ineptitude while causing their high-scoring American Basketball Association forefathers to cringe. The Nets and the San Antonio Spurs are crossing new frontiers of offensive ineptitude while embarrassing their high-scoring ABA forefathers. Please tell me if the sentences above mean the same. Yes.

Example2: The girl turned up late Sunday at a convenience store in East Palo Alto, about 30 miles from her home. The girl turned up late Sunday night at an East Palo Alto convenience store about 30 miles from her home. Please tell me if the sentences above mean the same. Yes.

Example3: "There were," said board member and Nobel-prize winning Stanford physicist Douglas Osheroff, "some extremely bad decisions." Board member Douglas Osheroff, a Nobel-prize winning Stanford physicist, said: "There were some extremely bad decisions." Please tell me if the sentences above mean the same. Yes.

Test Input Answer: Yes. ✗

**SeDPO**:
Example1: He was sentenced in June to more than seven years in prison for securities fraud, perjury and other crimes. He was sentenced to more than seven years in prison after pleading guilty to charges including securities fraud. If the first sentence is true, is the second one also true? No.

Example2: He really left us with a smile on his face and no last words, daughter Linda Hope said. "He really left us with a smile on his face and no last words...He gave us each a kiss and that was it." she said. If the first sentence is true, is the second one also true? No.

Example3: This Palm OS smart phone is the last product the company will release before it becomes a part of palmOne. This was almost certainly its last full quarter before the company becomes a part of Palm. If the first sentence is true, is the second one also true? No.

Test Input Answer: No. ✓

**Task: Paraphrase (QQP)**
Test Input: What is teeth scaling? How is teeth scaling done? Do those questions have the same meaning?

**UPRISE**:
Example1: How much time is needed to master Quantum Mechanics from scratch? How much time is required to master Quantum mechanics from scratch? Do those questions have the same meaning? Yes.

Example2: Why was the holocaust so slow? Why did the holocaust happen so slowly? Do those questions have the same meaning? Yes.

Example3: Why can't we make artificial blood? Why we can't make artificial blood? Do those questions have the same meaning? Yes.

Test Input Answer: Yes. ✗

**Se²**:

Example1: The best of 2016: Which are the best Bollywood movies in 2016? Which Bollywood movie you like the most in 2016? Do those questions have the same meaning? Yes.

Example2: How does starving help to lose weight? How can starving yourself to lose weight adversely affect your health? Do those questions have the same meaning? Yes.

Example3: Why are so few drugs with promising animal trials tested in humans? How much do clinical trials for drugs cost? Do those questions have the same meaning? No.

Test Input Answer: Yes. ✗

**SeDPO**:

Example1: Is PC gaming better than console gaming? Is PC gaming better? Please tell me if those questions are the same. Yes.

Example2: Should people over 95 not be allowed to vote? Should people over 93 not be allowed to vote? Please tell me if those questions are the same. Yes.

Example3: Which hp laptop is best for a graphic design/gamer? Which is best HP or Dell laptop? Would you say that these questions are the same? No.

Test Input Answer: No. ✓

**Task: Paraphrase (PAWS)**

Test Input: Do these mean the same? Wilbur was born on 1 March 1921 in North Caldwell, New Jersey and grew up in New York City. Wilbur was born in North Caldwell, New Jersey March 1, 1921, and grew up in New York City.

**UPRISE**:

Example1: Are these paraphrases? Born in Gosforth, Northumberland, he moved to the south as a boy to Wiseton Estate, near Retford, Nottinghamshire, when his father found jobs there. Born in Retford, Nottinghamshire, he moved as a boy to Wiseton Estate, near Gosforth, Northumberland, when his father found jobs there. No.

Example2: Do these mean the same? J.David Spurlock was born on November 18, 1959 in Memphis, Tennessee. He moved to Dallas, Texas in 1973. David Spurlock was born on 18 November 1959 in Dallas, Texas, and moved to Memphis, Tennessee in 1973. No.

Example3: Do these mean the same? Joe was born in Somerville, Massachusetts on March 27, 1929 and grew up in Quincy, Massachusetts. Joe was born on March 27, 1929 in Quincy, Massachusetts, where he grew up in Somerville, Massachusetts. No.

Test Input Answer: NO. ✗

**Se²**:

Example1: Are these paraphrases? The following sound changes from Proto-Celtic to Welsh, Cornish and Breton are summarised in the regular consonant table. The regular consonantal sound changes from Proto-Celtic to Welsh, Cornish and Breton are summarised in the following table. No.

Example2: Do these two sentences from wikipedia have the same meaning? Kennell was born in Colorado Springs, Colorado, and spent her early years between the Rockies and Dunedin, Florida. Kennell was born in Dunedin, Florida, and spent her early years between the Rockies and Colorado Springs, Colorado. No.

Example3: Do these two sentences from wikipedia have the same meaning? Robert Maass was born in East Orange, New Jersey, to study German immigrants Hedwig and Clara Maass. Robert Maass was born in East Orange, New Jersey, to German immigrants Hedwig and Clara Maass. Yes.

Test Input Answer: No. ✗

**SeDPO**:

Example1: Do these two sentences from wikipedia have the same meaning? Several of these names were chosen to correspond to their international equivalents in rough chess, and not as literal translations of the Japanese names. These names were chosen to correspond to their international counterparts in the rough chess and not as literal translations of Japanese names. Yes.

Example2: Do these two sentences from wikipedia have the same meaning? Due to the results obtained in the previous round, Kevin Gleason received + 30kg , Gianni Morbidelli + 20kg and Pepe Oriola + 10kg. Due to the results of the previous round Kevin Gleason received + 30kg, Gianni Morbidelli + 20kg and Pepe Oriola + 10kg. Yes.

Example3: Do these two sentences from wikipedia have the same meaning? When combined for joint or coalition operations, it was known as a common or employed air operations centre for coalition operations. When combined for joint or coalition operations, it was known as a joint or employed air operations center for coalition operations. Yes.

Test Input Answer: Yes. ✓

