# OpenReview forum: "Learning to Rank for In-Context Example Retrieval"
_NeurIPS.cc/2025/Conference — NeurIPS 2025 poster_

### Official Review · Reviewer_ESxb · 2025-06-07

**Clarity:** 3
**Significance:** 3
**Originality:** 3
**Rating:** 4
**Confidence:** 4

**Summary:**

The paper considers the problem of selecting in-context learning examples to improve a language model's performance.
It formulates in-context example (ICE) retrieval as a learning-to-rank problem and uses preference orderings, rather than classification.

Concretely, given a prompt $x$, the goal is to select examples $e_1, e_2, \dots, e_K$ from a corpus and add them to a problem.
It trains a retriever that selects the top $w$ examples in one batch given the currently selected examples.
It repeats this operation until $K$ examples are selected.

It uses the likelihood of the correct response given a sequence of examples as reward signals.
To generate preference labels, it compares the likelihoods of different ICE sequences.
It trains the retriever using a variation of DPO, to prefer sequences that lead to higher rewards.

Empirically, the method outperforms prior retrieval-based ICL methods on 9 NLP tasks.

**Questions:**

Some questions are in the weaknesses section above.

As state-of-art language models support much longer context lengths, does this generalize to larger number of examples (allowing selecting more than $K$ examples during inference)?

**Ethical Concerns:**

["NO or VERY MINOR ethics concerns only"]

**Final Justification:**

The authors have clarified the contributions and baseline method selection. It shows convincing results that ranking-based ICE selection outperforms score-based ICE selection. However, as other reviewers noted, the results are only evaluated on multi-choice question benchmarks, which makes the contributions relatively limited.

I have adjusted the score to 4.

**Limitations:**

The limitations are discussed in Section .9 in the appendix.

**Quality:**

3

**Strengths And Weaknesses:**

### Strengths

**Novelty.**
Using a DPO-based pairwise loss to train a retriever is novel and inspirational.
DPO is known to be stable and is a reasonable choice for training a retriever to achieve better model outputs.

**Empirical performance.**
The method is empirically evaluated in multiple NLP domains and outperforms multiple baseline algorithms, including Se2 and UPRISE.

### Weaknesses

**Missing baselines in empirical evaluation.**
One existing work that is closely relevant is Active Example Selection ([31] in the paper).
The paper claims that this method "suffer from training instability" (line 63),
and "(not) considers ranking orders of examples across prompts" (line 66).

If I understand this work correctly, Active Example Selection uses an RL approach to select examples one by one.
So it does consider the order of examples.
Also, this baseline algorithm is not evaluated empirically, so it's unclear if the proposed algorithm is indeed advantageous.

---

> ### Author Rebuttal · Authors · 2025-07-27
>
> Thank you very much for your time and effort in reviewing our work. We would like to address some of the comments that may stem from misunderstandings. If you find our response helpful, we would sincerely appreciate it if you could consider raising your score :)
>
> **W1/W2. Clarification of our contributions:**
> As described in both the introduction and Figure 1, our key contribution lies in modeling the ranking among ICEs—for example, comparing and ranking different K-shot prompts. In contrast, the Active Example Selection (AES) method (citation [31]) is not closely-related to our work. AES focuses on sequentially selecting examples to construct a K-shot prompt, emphasizing the internal ordering within ICEs. Our baseline, Se2, is the most advanced follow-up to AES, following the same paradigm for intra-ICE ordering but without the need for active learning. The following shows the results of AES on related tasks  (taken from Se2 [13] in paper).
>
> | Method | PAWS (ACC) | AGNews (ACC) | MNLI-m (ACC) | MNLI-mm (ACC) |
> |---|---|---|---|---|
> | AWS  | 51.7 | 78.2 | 43.2 | 29.5 |
> | SE-DPO | 73.0 | 90.7 | 70.6 | 72.0 |
>
> Considering both relevance and performance, AES is not suitable as our baseline. We will highlight the evidence in the related work section.
>
>
> **Q. Clarification on inference results:**
> We have provided inference experiments with $K$ ranging from 1 to 15, using various LLMs, as shown in Table 4. A detailed analysis of these results can be found in lines 268–276 of the paper. Our empirical study (Table 3) demonstrates that SEDPO can generalize to a larger number of examples.

---

> ### Comment · Reviewer_ESxb · 2025-08-04
>
> Thanks for the additional results and the clarifications. I see that Se2 is a strong baseline to compare SE-DPO with.
>
> Some additional questions:
>
> What is AWS method in the table in your response? Is that the result for AES?
>
> More broadly, does this work suggest that ranking relationship between ICEs is easier to learn than predicting their absolute scores?

---

> ### Author Response · Authors · 2025-08-05
>
> Thank you for taking the time to review our paper and offer constructive feedback!
>
> **What is AWS method in the table in your response? Is that the result for AES?**
> > We apologize for this typo, AWS method in the table is the result for AES (taken from Se2 [13] in paper).
>
>
> **Does this work suggest that ranking relationship between ICEs is easier to learn than predicting their absolute scores?**
> > Yes. As we discussed in lines 22 to 28, existing methods train retrievers as classifiers using absolute scores, but inference relies on ranking, creating a goal mismatch. Learning to rank is more consistent with the test scenario, and we also provide empirical evidence in lines 223 to 230 of the paper.

---

> ### Comment · Reviewer_ESxb · 2025-08-05
>
> Thanks for the clarifications. I have updated the score to 4 and added justification above.

---

> > ### Author Response · Authors · 2025-08-06
> >
> > Thank you very much for your understanding!

---

### Official Review · Reviewer_Cqhk · 2025-06-23

**Clarity:** 3
**Significance:** 3
**Originality:** 2
**Rating:** 4
**Confidence:** 3

**Summary:**

This paper proposes an approach to selecting and ranking examples for ICL. The approach is based on a ranking formulation based on DPO and sequentially choosing examples. The paper adapts DPO to this setting and uses a relaxation to derive a tractable algorithm. Experiments on multiple choice questions (MCQ) show SOTA performance compared to several baselines.

**Questions:**

* Eq (2): this is defined for a small \mathcal{Y}, as in your MCQ setting. How can \phi_{LLM} be defined for general outputs? How about sampling y's from the model for \mathcal{Y}?
* Line 173: can you include a proof for transitive closure?
* Can you comment on the cost of constructing the training data?
* How does performance vary with different values of T?

Please detail relationship to these related papers:
* https://arxiv.org/pdf/2503.08030
* https://www.arxiv.org/pdf/2506.08607 (made public after NeurIPS submission)
* https://www.arxiv.org/pdf/2506.04579 (made public after NeurIPS submission)

Minor / Typos
* SE-DPO appears in lines 6 and 34, but defined only in line 39.
* Eq (1): why are examples ordered from K to 1? This is a bit strange.
* Line 90: the notation E_e(e) and E_x(x) is confusing. Consider something like E_{example}(e) and E_{input}(x).
* Line 92: “we first encodes and indexes” => “we first encode and index”
* Line 125: The notation \tilde{x}_1\in\mathcal{C} is confusing since \mathcal{C} is a corpus of (x,y) examples (line 83) while x represents inputs only.
* Line 146: “we denotes” => “we denote”

**Ethical Concerns:**

["NO or VERY MINOR ethics concerns only"]

**Final Justification:**

After reading through the authors response and the other reviews, I would like to keep my "borderline accept" recommendation.

The authors did a good job in the rebuttal addressing some of the questions. The additional experimental results on scalability to larger models and the effect of the number of compared examples T are a great addition and are helpful in understanding the performance of the proposed approach. The authors also clearly differentiated their work from other previous and concurrent work: their focus on formulating the retriever training as a direct learning-to-rank problem with a pairwise preference objective distinguishes their work from others. The authors also adequately addressed my original concerns about computational cost of training data construction.

I think that the main limitation of the paper is its restricted application to MCQ. In their rebuttal, the authors confirmed that their attempts to apply the method to open-ended question answering were unsatisfactory. Furthermore, novelty seems to be OK but not outstanding: the core idea is applying DPO to the problem of in-context example retrieval. While the application is novel and the engineering to make it work is sound, it mostly cleverly combines existing components. The work is high-quality, but it builds upon established paradigms.

If there was an option for a 4.5 score I would choose that, but don't feel it's a 5 paper.

**Limitations:**

See weaknesses/questions above.

**Paper Formatting Concerns:**

No concerns

**Quality:**

3

**Strengths And Weaknesses:**

Strengths:
* Identifying mismatch between train and test in ICL, and using a ranking formulation for training to resolve it.
* The empirical evaluation seems comprehensive and well executed.
* The empirical results show SOTA performance compared to strong baselines.

Weaknesses:
* Generalizability beyond MCQ.
* Potentially computationally expensive dataset construction.

---

> ### Author Rebuttal · Authors · 2025-07-27
>
> We are grateful for your thoughtful suggestions and will incorporate additional details into the paper accordingly. If you find our responses helpful, we would sincerely appreciate it if you could consider raising your score :)
>
> **Q1. On general output and the size of $\mathcal{Y}$:**
> If the general output refers to open-ended QA, the set $\mathcal{Y}$ becomes impractically large, making Equation (2) intractable to compute. We have experimented with sampling $y$ values for $\mathcal{Y}$ directly from the model; however, the results were unsatisfactory. Unlike MCQs, open-ended QA lacks a clear way to quantify and normalize the quality gap between good and bad answers, making reliable supervision difficult.
>
>
> **Q2. Proof of transitivity:**
> By the definition of $\succ$ (lines 117-118):
> - If $e^a \succ e^b|x$, then for all $y \in {\mathcal{Y}}\_{\text{gt}}$ , $\mathrm{S}\_\mathrm{MCQ}(e^a \oplus x,y) > \mathrm{S}\_\mathrm{MCQ}(e^b \oplus x,y)$. (1)
> - If $e^b \succ e^c|x$, then for all $y \in \mathcal{Y}\_{\text{gt}}$, $\mathrm{S}\_\mathrm{MCQ}(e^b \oplus x,y) > \mathrm{S}\_\mathrm{MCQ}(e^c \oplus x,y)$. (2)
>
> Note that $\mathrm{S}\_\mathrm{MCQ}(\cdot)$ is a scalar-valued function, and its outputs are real numbers. The ">" relation on the real numbers is transitive: for any real numbers $a, b, c$, if $a > b$ and $b > c$, then $a > c$.
>
> Applying this transitivity to (1) and (2) for each $y \in \mathcal{Y}_{\text{gt}}$:
> For all $y \in \mathcal{Y}\_{\text{gt}}$, $\mathrm{S}\_\mathrm{MCQ}(e^a \oplus x,y) > \mathrm{S}\_\mathrm{MCQ}(e^b \oplus x,y)$ and $\mathrm{S}\_\mathrm{MCQ}(e^b \oplus x,y) > \mathrm{S}\_\mathrm{MCQ}(e^c \oplus x,y)$ implies $\mathrm{S}\_\mathrm{MCQ}(e^a \oplus x,y) > \mathrm{S}\_\mathrm{MCQ}(e^c \oplus x,y)$. (3)
>
> By the definition of $\succ$ again, (3) implies $e^a \succ e^c|x$.
> Thus, the relation $\succ$ is transitive.
> Q.E.D.
>
>
> **Q3. Cost of constructing training data:**
> We provide below the average cost of constructing preference data for all tasks. Experiments were conducted on 8 V100-32GB GPUs with a scoring batch size of 10, using GPT-Neo-2.7B as the ICL model. For each $x$, we only sample $T = 20$ preference pairs. This means *the number of processed preference pairs is less than the number of scored entries*. For instance, on NLI, the total time for constructing scored data is 794× greater than that for preference data:
>
> | Steps to construct training data | Speed |
> |--------|---------|
> | Construction of scored data (Se2) | 0.0241 s/entry |
> | Constructing preference data from (SEDPO) | 0.0007 s/pair |
>
>
> **Q4.**
> We provide below the extra ablation results on Paraphrase for different $T$, following the setup of Table 2.
>
>
> | Method | T=1 | T=10 | T=20 | T=30 |
> |--------|---------|---------|---------|---------|
> |SE-DPO (β=0.02) | 66.1 | 74.5 |  77.9 | 79.9 |
>
> As T increases, the performance of SEDPO improves because a broader preference ranking is learned. However, when T is too large, the performance gain decreases due to more low-confidence LLM rankings. As we have discussed this in lines 131-136 and lines 241-243, low-confidence LLM rankings can negatively affect learning --- there is a trade-off.
>
> **Minor/Typos.** Thank you for your careful reading. We will thoroughly correct all typos and minor issues in the revised version.

---

> > ### Comment · Reviewer_Cqhk · 2025-08-04
> >
> > Thanks for the clarifications.
> > A few follow up questions.
> >
> > 1. You noted in your rebuttal that applying SE-DPO to open-ended QA was "unsatisfactory" due to the difficulty in normalizing answer quality. Could you elaborate on the specific failure modes you observed?
> >
> > 2. Can you please also add details on overlapping contributions or key differentiators to these recent papers:
> > * https://arxiv.org/pdf/2503.08030
> > * https://www.arxiv.org/pdf/2506.08607 (made public after NeurIPS submission)
> > * https://www.arxiv.org/pdf/2506.04579 (made public after NeurIPS submission)
> >
> > 3. As noted by other reviewers, the performance gains over the baselines shrink as the model size increases. While you position this as a strength for smaller models, what is your hypothesis for why these gains diminish? Are there any modifications which may unlock more significant gains on state-of-the-art models?
> >
> > Thanks.

---

> > > ### Author Response · Authors · 2025-08-05
> > >
> > > We deeply appreciate your commitment to reviewing our work and offering meaningful feedback to enhance this paper!
> > > We will certainly add the discussion presented in our rebuttal to the revised version.
> > >
> > > **Discussion for specific failure mode in open-ended QA.**
> > > > We have experimented with sampling ${y'}$  from the model for $\mathcal{Y}=(y_{\text{ref}},y'_1, y'_2,...,y'_d)$, the result is close to zero-shot. We examined the cases and found that the sampled ${y'}$ is not necessarily an incorrect answer; it may even be better than the reference answer, which leads to low confidence in the reward signal.
> > >
> > > **Add details on overlapping contributions or key differentiators to these recent papers.**
> > >
> > > We will cite these recent papers [1,2,3] in the related work and include a brief discussion in the revised version.
> > > - [1] Xiang Gao et al. Learning to Search Effective Example Sequences for In-Context Learning
> > > - [2] Kiran Purohit et al. Sample Efficient Demonstration Selection for In-Context Learning
> > > - [3] Jianfei Zhang et al. Selecting Demonstrations for Many-Shot In-Context Learning via Gradient Matching
> > >
> > > > SEDPO vs. BESC [1]
> > > > - **Key Difference:** BESC considers the inner order of the ICEs within each prompt, using a contrastive loss to learn how to incrementally construct the best sequence with dynamical length step-by-step; SEDPO emphasizes the rankings between different prompts.
> > > > - **Overlapping:** Learning from LLM Feedback, aware of sequential dependence of the ICEs within each prompt, tackling computational complexity of original problem.
> > >
> > > >  SEDPO vs. CASE [2]
> > > > - **Key Difference:** CASE prioritizes the efficiency of the selection process, framing selection as a "top-m arm identification" problem with absolute training reward; SEDPO prioritizes the quality of retrieved examples, framing selection as a "learning to rank" problem with pairwise training reward.
> > > > - **Overlapping:** Learning from LLM Feedback, acknowledge the high cost of LLM inference.
> > >
> > > >  SEDPO vs. CLG [3]
> > > > - **Key Difference:** CLG is a task-level selection method in many-shot scenarios, where scalability is essential and CLG selects a fixed set of examples for all of the test queries through the gradient matching technique. SEDPO is optimized for few-shot scenarios where the query-specific precision of each example is critical.
> > > > - **Overlapping:** Learning from LLM Feedback
> > >
> > >
> > >
> > > **Hypothesis for gains diminish as the model size increases and modifications for unlocking**
> > > > Thank you for pointing this out. The retriever used for Table 4 was trained on GPT-Neo-2.7B, and its gains diminish as the model size increases --- We attribute this to the quality of retrieved examples being limited by the LLM that provides the ranking. For a quick verification, we train another retriever model on our smallest task WSC using preference data constructed with Llama3-8B-Instruct, and its inference results on Llama3-8B-Instruct are as follows. An 8.65% relative improvement in 3-shot settings demonstrates the effectiveness of these modifications.
> > > | Method | WSC (ACC)|
> > > |---|---|
> > > | Se2 | 53.85 |
> > > | SeDPO | 62.50 |
> > > >
> > > > We have also noted that the recent papers you recommended provide new insights into ICL for large models, representing an important direction for future work. We believe that combining SEDPO with task-level selection methods like CLG [3], thereby enabling the constructed fixed example set to capture diverse preferences, is expected to more elegantly address the issue that SEDPO requires retraining for scaling.

---

### Official Review · Reviewer_TQpj · 2025-06-29

**Clarity:** 1
**Significance:** 2
**Originality:** 2
**Rating:** 3
**Confidence:** 4

**Summary:**

This paper proposes a ranking retrieval method, where the preference rankings between ICEs are given by comparing the likelihood of the LLM generating the correct answer conditioned on each exemplar.

**Questions:**

1. Could you explain more on Eq. (3)? As currently written, $p(rank)=\frac{exp(-rank)}{exp(-1)+exp(-2)+..+exp(-L)}$, appears to be a fixed distribution over ranks, independent of the specific sample $e_c$, assuming $L$ is fixed. There are no sample-dependent variables in the equation, which makes it unclear how this relates to individual examples.
Is this meant to represent a normalized ranking distribution averaged over multiple sampling rounds for each $e_c$? Clarifying this would improve the reader's understanding.

2. In line 152, the formulation of $Z(x)$ is not defined. It would be helpful to either include its explicit definition or provide a citation that explains it, to enhance clarity and reproducibility.

**Ethical Concerns:**

["NO or VERY MINOR ethics concerns only"]

**Final Justification:**

I would like to raise my score to 3.

**Limitations:**

This paper has discussed the limitations.

**Paper Formatting Concerns:**

No formatting concerns.

**Quality:**

2

**Strengths And Weaknesses:**

Strength:

1. The proposed method achieves significant improvement by adopting GPT-Neo-2.7B.
2. A wide range of datasets is adopted during the evaluation.

Weakness:

1. The main motivation of the paper is based on the claim that "dominant approaches train the retriever as a classifier" (line 22) and that "learning-to-rank (LTR) remains underexplored in in-context learning (ICL)" (lines 29–32). However, this motivation is not fully convincing, as several prominent ICL methods—such as UDR [1], PromptPG [2], and LLM-R [3]—already employ ranking-based retrievers, where samples with the highest ranking scores are selected. As such, the paper’s premise that LTR is underexplored appears overstated. Furthermore, the proposed method does not present a substantial novelty over these existing approaches, and the experimental section lacks a direct comparison with other ranking-based ICL methods (only UDR is included).

[1] Xiaonan Li, Kai Lv, Hang Yan, Tianyang Lin, Wei Zhu, Yuan Ni, Guotong Xie, Xiaoling Wang, and Xipeng Qiu. Unified demonstration retriever for in-context learning.

[2] Lu P, Qiu L, Chang K W, et al. Dynamic prompt learning via policy gradient for semi-structured mathematical reasoning[J]. ICLR 2023.

[3] Wang L, Yang N, Wei F. Learning to retrieve in-context examples for large language models[J]. arXiv preprint arXiv:2307.07164, 2023.

2. According to Table 4, the performance gains of the proposed method diminish as the model size increases. Specifically, the improvements for LLaMA3-8B and LLaMA3.3-70B are relatively modest—only 1.2% and 0.7%, respectively. This suggests that the proposed method may not generalize well to larger language models, raising questions about its scalability and broader applicability.

3. Some of the equations or notations are not clearly explained, which confuses the readers. Please refer to the questions for details.

4. This paper emphasizes its contribution over the rankings between ICEs. Will this paper also consider the inner order of the ICEs within each prompt? According to the implementation details (line 185), the shot number $K$ is only 3. Given such a small number, does the inner ordering of ICEs meaningfully affect performance? Clarifying whether and how this aspect was considered would strengthen the experimental analysis.

---

> ### Author Rebuttal · Authors · 2025-07-27
>
> Thank you very much for your time and effort in reviewing our work. We sincerely appreciate your thoughtful comments and would like to address the concerns raised. If you find our responses helpful, we would be truly grateful if you could consider adjusting your score accordingly :)
>
>
> **W1. Clarifying our contributions.** We believe there may be some misunderstandings regarding the uniqueness of our work. As described in both the introduction and Figure 1, our study focuses on the ranking formulation among ICEs (i.e., different $K$-shot prompts), which is a central contribution highlighted in both the introduction and Figure 1. In contrast, prior works primarily adopt classification-based approaches:
> - **PromptPG** focuses on classification-based retrieval and does not explicitly investigate the internal or external ordering of ICEs. While it notes that ordering can affect model performance, this is mentioned only in the analysis section and not as a core research objective.
> - **LLM-R** similarly does not study the preference order among ICEs. The “ranking” mentioned in its analysis refers to a screening mechanism within a classification formulation.
> - **UDR** introduces a local ranking regularizer into the classification formulation, but still fundamentally operates in a classification framework. We have provided a detailed analysis of UDR’s limitations in lines 68, 203–205, and 218–220, along with supporting experiments in Tables 1 and 2.
>
>
> **W2. Clarification on Table 4 and additional experiments.** Table 4 in our paper aims to demonstrate that the performance improvement of our proposed method becomes more pronounced as the model size decreases (lines 274–276). This experiment is designed to explore the boundary of transferability and to help identify suitable application scenarios for SEDPO. We would like to respectfully note that smaller models remain a highly active research area. To further support this point, we provide below an additional evaluation on **GPT-2 XL (1.5B)**, following the same setup as in Table 3:
>
> | Method | 1-shot |3-shot |6-shot |9-shot |12-shot |15-shot |Avg. |
> |--------|---------|---------|---------|---------|---------|---------|---------|
> | UPRISE | 69.15 | 69.41 | 69 .84 | 69.82 | 69.97 | 70.16 | 69.72 |
> | BM25 | 57.61 | 58.47 | 58.81 | 58.67 | 59.31 | 60.06 | 58.82 |
> | SBERT | 57.93 | 57.52 | 58.98 | 59.56 | 58.57 | 58.29 | 58.48 |
> | Se2 | 73.90 | 72.89 | 72.90 | 72.80 | 72.80 | 72.70 | 73.00 |
> | SEDPO| 75.03 | 78.93 | 79.46 | 79.17 | 78.99 | 79.19 | 78.46 |
>
> SEDPO outperforms the second best by up to 5.46%, which demonstrates better transferability on smaller models. We hope this additional evidence further supports the generalizability and effectiveness of our method. If space allows, we’ll add more small-LLM results to Table 4.
>
> **W3.** Kindly refer to our responses to Q1 and Q2 for clarifications.
>
> **W4. Transferability on shot number and intra-ICE ordering.** Table 4 presents the transferability of different methods across 1 to 15-shot settings. While our primary focus is on the ordering between ICEs, we also analyze the effect of intra-ICE ordering through additional experiments. This analysis is provided in Table 3 and discussed in lines 259–262.
>
> **Q1. Greedy sampling strategy (Section 3.2)** We appreciate your attention to the selection process for constructing $K$-shot prompts $e_{[1:K]}$ for training. As described in Section 3.2, considering all non-repetitive combinations is computationally infeasible (line 122). Therefore, we use a greedy sampling approach guided by Eq. (3) to select each example $e_c$ within $e_{[1:K]}$. When selecting $e_1$, we randomly sample $L$ examples (Eq. (2)), rank them using the LLM score, and then sample from these candidates based on the probabilities given in Eq. (3). This approach, which takes into account the importance and diversity of examples, is expected to be more effective than uniform sampling, as cited in line 123.
>
>
> **Q2. Clarifying $Z(x)$ in Eq. (5).**
> Thank you for raising this point. The partition function $Z(x)$ is introduced in Eq. (5), and reference of DPO \[4]—which provides a detailed explanation—is cited in the paragraph preceding the formula. To enhance clarity for readers, we will consider adding more explicit references near the $Z(x)$ itself.

---

> > ### Comment · Reviewer_TQpj · 2025-08-07
> >
> > Thank you for your clarification! Your response solved some of my concerns. However, I still find the contribution related to the ranking component somewhat limited when compared to prior work. Additionally, considering that larger LLMs can now achieve strong zero-shot performance—even surpassing the few-shot results of smaller models—I believe it is more important to focus on improvements for larger LLMs. The proposed method, however, primarily demonstrates gains on smaller models, which limits its broader impact.
> >
> > Based on the above reasons, I would like to raise my score to 3.

---

> ### Author Response · Authors · 2025-08-07
>
> Thank you for taking the time to read our manuscript and for your feedback. We’re glad that several of our clarifications addressed your concerns. Below, we respond in detail to your remaining points.
>
> **Reviewer’s concern: The contribution related to the ranking component seems limited compared to prior work.**
> > Our contribution lies in modeling retriever training as a ranking problem rather than a classification problem, which is consistent with the test scenario (lines 22-28). Prior work models retriever training as a classification problem, and the training objectives of the two are completely different.
> >
> > Moreover, the prior ranking within prompts and our ranking between prompts are two distinct concepts:
> > - **Between prompts** aims to determine which information is more useful, focusing on the different utility provided by different prompts.
> > - **Within prompts** adjusts the input order between selected examples, with more focus on the correct dependencies of the same information.
>
> > We sincerely hope that the reviewer can point out the specific limitations of our ranking component compared to prior work. We have carefully examined the papers you provided, and PromptPG and LLM-R are not ranking-based ICL methods; UDR is already our baseline, and we noted in the paper that it trains the retriever by learning classification rather than directly learning to rank.
>
> **Reviewer’s concern: With large LLMs already achieving strong zero-shot results, it is more important to focus on improvements for larger LLMs**
> > We acknowledge the zero-shot capabilities of larger LLMs, but few-shot learning can also benefits large models compared to the zeroshot. The decline in gains we are discussing is relative to the second-performed retriever, not the zero-shot (lines 274 - 276). We present below a comparison between our method and the zero-shot results on a larger model, where the gains remain significant.
> | Model | 0-shot | SE-DPO (1-shot) | SE-DPO (3-shot) | SE-DPO (6-shot) | SE-DPO (9-shot) | SE-DPO (12-shot) | SE-DPO (15-shot) |
> |---|---|---|---|---|---|---|---|
> | Llama3-8B-Instruct | 56.4 | 71.9 (+15.5%) | 77.4 (+21%) | 78.5 (+22.1%) | 79.3 (+22.9%) | 80.2 (+23.8%) | 80.3 (+23.9%) |
> | Llama3.3-70B | 67.6 |  78.6 (+11.0%) | 81.0 (+13.4%) | 82.3 (+14.7%) | 82.9 (+15.3%) | 83.2 (+15.6%) | 83.2 (+15.6%) |
> >
> > In discussion with Reviewer Cqhk, we also explored retraining our retriever under the supervision of rankings from the larger LLM to further make our method outperform the baseline retriever by a large margin.
>
> > Finally, the field of LLMs has not yet reached a consensus on focusing solely on improving zero-shot performance by scaling model size, given that the deployment and inference costs of large LLMs are high, and agent-based features such as deep research also rely on RAG systems. While different scholars adhere to different propositions, we believe these debates should not undermine the validity of our current contributions.
>
> Please let us know if any phrasing still seems unclear or if you’d like additional evidence or discussion on any of these points. Thank you again for your feedback!

---

### Official Review · Reviewer_R6FR · 2025-07-02

**Clarity:** 3
**Significance:** 3
**Originality:** 3
**Rating:** 4
**Confidence:** 4

**Summary:**

This paper introduces **SE-DPO**, a novel learning-to-rank algorithm for in-context example (ICE) retrieval in large language models (LLMs). Unlike classification-based methods, SE-DPO aligns retrieval scores with LLM preferences by optimizing pairwise rankings of ICEs, leveraging Direct Preference Optimization (DPO) with sequential relaxation. Experiments across 9 NLP tasks show SE-DPO outperforms state-of-the-art baselines by up to 18%, particularly in challenging tasks like PAWS and MultiRC. Ablation studies confirm the importance of ranking formulation and diversity in ICE selection. The method also demonstrates strong transferability across model sizes and shot numbers. Limitations include reliance on dominant preferences and potential biases, addressed via safeguards discussed in the broader impacts section.

**Questions:**

1. The impact of different embedding models on the results. I am not clear about whether the influence of embeddings on the results is good or bad. This paper uses "BERT-base-uncased" as the encoder model, but nowadays there are many better text embedding models available.
2. Preference Labeling Stability: The paper constructs ranked data by comparing LLM likelihoods, but how sensitive is SE-DPO to noise or inconsistencies in these scores? If the LLM’s confidence in ranking ICEs is low (e.g., near-tie scores), does the method degrade?

**Ethical Concerns:**

["NO or VERY MINOR ethics concerns only"]

**Final Justification:**

I appreciate the authors' response and would like to keep my positive score.

**Limitations:**

Yes

**Paper Formatting Concerns:**

No.

**Quality:**

3

**Strengths And Weaknesses:**

## Strengths:
1. **Effective Ranking Formulation:** SE-DPO outperforms classification-based methods by explicitly optimizing pairwise rankings of in-context examples (ICEs), leading to significant performance gains (up to 18%) across multiple NLP tasks.
2. **Strong Generalization:** The method demonstrates robust transferability across different LLM sizes and shot numbers, showing consistent improvements even when tested on unseen inference settings.

## Weaknesses:
1. **Computational Overhead:** The sequential ranking process and preference alignment introduce additional complexity, making training more resource-intensive compared to simpler classification-based retrievers.
2. **Bias and Fairness Risks:** Like other retrieval-based methods, SE-DPO may inherit biases from LLM-generated preferences, potentially reinforcing harmful or misleading examples without careful mitigation.

---

> ### Author Rebuttal · Authors · 2025-07-27
>
> Thank you very much for your constructive feedback. We sincerely appreciate your time and effort in reviewing our work and would like to address all of your concerns. If you find our responses helpful, we would be truly grateful if you could consider adjusting your rating accordingly :)
>
> **W1/W2. Fairness, overhead, and discussion clarity.**
> We have discussed several limitations of our method—including fairness—in the appendix. Since our algorithm does not introduce inference-time overhead, the additional computational cost remains affordable. We will revise the conclusion section to better highlight these points.
>
>
> **Q1: Text encoder fairness and ablation results (lines 183–188).**
> As described in the paper, both SEDPO and all trainable baselines use BERT-base-uncased to ensure fair comparison (lines 183–188, Table 2).
> To further show the influence of the embedding model, we conducted additional ablation experiments with RoBERTa-base:
>
> | Encoder | Paraphrase-Avg |  Paraphrase-MRPC (ACC/F1) | Paraphrase-PAWS (ACC) | Paraphrase-QQP (ACC/F1) |
> |--------|---------|---------|---------|---------|
> | BERT-base-uncased |  77.9  | 77.9/85.6 | 73.0 | 77.6/75.0 |
> | RoBERTa-base | 85.7 |  86.3/90.5 | 88.1 | 83.6/ 80.0 |
>
> RoBERTa-base is known to outperform BERT-base-uncased across a range of NLP tasks, and as shown above, SEDPO benefits from stronger embeddings as expected. We will include these results in Table 2.
>
>
> **Q2: Robustness to noisy preferences (lines 134 and 242, Table 2).**
> As shown in Table 2 and discussed in lines 134 and 242, replacing our original preference data with randomly sampled preference pairs of the same size significantly degrades performance. This effect is more pronounced than the degradation seen in cases of close-score (low-confidence) preferences. These results suggest that SEDPO is sensitive to preference noise, and that low-confidence LLM rankings can negatively affect learning. We will revise the relevant sections to make this insight clearer and more prominent.

---

> > ### Comment · Reviewer_R6FR · 2025-08-04
> > **Thanks for the response.**
> >
> > I'd like to keep my positive score.

---

> > > ### Author Response · Authors · 2025-08-06
> > >
> > > Thank you for your time and effort in reviewing our paper !

---

### Note · Authors · 2025-08-12

We sincerely thank the reviewers and the AC for their efforts throughout the review process. As some of our responses did not receive explicit feedback, we briefly summarize the key clarifications below and respectfully invite further consideration.

**Clarifications regarding the novelty of our ranking-based retriever**
> In this paper we characterize retrievers as classification-based or ranking-based according to their training objective, not by whether "ranking" is mentioned in their paper. The training objective determines whether the model learns classification or ranking behavior. For example, some prior works construct training data using a ranking-style procedure, but their optimization remains a classification objective; therefore, they should not be regarded as ranking-based retrievers. This is why we say "learning-to-rank remains underexplored in in-context learning (ICL)" rather than "ranking component is underexplored in ICL", and we have carefully examined the relevant works.

**Clarifications regarding the transferability to scaled LLMs.**
> Our paper results show retrievers trained on specific LLM feedback benefit smaller LLMs more than larger ones (analogous to observations in knowledge distillation), identifying where our proposal is most cost-effective. In addition, the learning-based retrievers show **significant improvements compared to zero-shot on larger LLMs**, indicating that ICL still works as inference LLM scales (see response to Reviewer TQpj). As inference LLM scales, our ranking-based retriever's gain over the classification-based one decreases, but can be widened by increasing the training cost and retraining the retriever on the preference data given by scaled LLM (see response to Reviewer Cqhk).

Once again, thank you for your careful consideration.

---

### Decision · Program_Chairs · 2025-09-17

**Decision:**

Accept (poster)

**Comment:**

I recommend accepting this paper, "Learning to Rank for In-Context Example Retrieval." The authors propose SE-DPO, a novel algorithm that trains a retrieval model using a ranking formulation rather than classification, where preference rankings between in-context examples (ICEs) are determined by comparing LLM likelihoods of generating correct answers. The approach outperforms state-of-the-art baselines across 9 NLP tasks, with particularly strong results for smaller models. While the performance gains diminish somewhat for larger LLMs, the authors have demonstrated that few-shot learning with their method still significantly outperforms zero-shot approaches, even for larger models. The reviewers noted the technical soundness and empirical effectiveness of the approach, and the authors provided thorough responses addressing concerns about novelty, transferability, and computational requirements. This work makes a valuable contribution to improving in-context learning through more effective example selection.